Corrected: Author correction

# TRPV1 SUMOylation regulates nociceptive signaling in models of inflammatory pain

Yan Wang[1], Yingwei Gao[1], Quan Tian[2], Qi Deng[1], Yangbo Wang[1], Tian Zhou[1], Qiang Liu[2], Kaidi Mei[2], Yingping Wang[1], Huiqing Liu[1], Ruining Ma[1], Yuqiang Ding[3], Weifang Rong[4], Jinke Cheng[1], Jing Yao[2], Tian-Le Xu[4], Michael X. Zhu[5] & Yong Li[1]

Although TRPV1 channels represent a key player of noxious heat sensation, the precise mechanisms for thermal hyperalgesia remain unknown. We report here that conditional knockout of deSUMOylation enzyme, SENP1, in mouse dorsal root ganglion (DRG) neurons exacerbated thermal hyperalgesia in both carrageenan- and Complete Freund's adjuvant-induced inflammation models. TRPV1 is SUMOylated at a C-terminal Lys residue (K822), which specifically enhances the channel sensitivity to stimulation by heat, but not capsaicin, protons or voltage. TRPV1 SUMOylation is decreased by SENP1 but upregulated upon peripheral inflammation. More importantly, the reduced ability of *TRPV1* knockout mice to develop inflammatory thermal hyperalgesia was rescued by viral infection of lumbar 3/4 DRG neurons of wild-type TRPV1, but not its SUMOylation-deficient mutant, K822R. These data suggest that TRPV1 SUMOylation is essential for the development of inflammatory thermal hyperalgesia, through a mechanism that involves sensitization of the channel response specifically to thermal stimulation.

[1] Department of Biochemistry and Molecular Cell Biology, Shanghai Key Laboratory for Tumor Microenvironment and Inflammation, Institute of Medical Sciences, Shanghai Jiao Tong University School of Medicine, Shanghai 200025, China. [2] Hubei Key Laboratory of Cell Homeostasis, College of Life Sciences, Wuhan University, Wuhan, Hubei 430072, China. [3] Department of Anatomy and Neurobiology, Collaborative Innovation Center for Brain Science, Tongji University School of Medicine, 200092 Shanghai, China. [4] Department of Anatomy and Physiology, Shanghai Jiao Tong University School of Medicine, 200025 Shanghai, China. [5] Department of Integrative Biology and Pharmacology, McGovern Medical School, The University of Texas Health Science Center at Houston, Houston, TX 77030, USA. These authors contributed equally: Yan Wang, Yingwei Gao, Quan Tian. Correspondence and requests for materials should be addressed to J.Y. (email: jyao@whu.edu.cn) or to Y.L. (email: liyong68@shsmu.edu.cn)

TRPV1 is a ligand-gated non-selective cation channel prominently expressed in small- and medium-diameter sensory neurons within dorsal root ganglia (DRG), where it functions as a sensor for an array of exogenous and endogenous chemical and physical stimuli, including capsaicin, noxious heat (>43 °C), and acidosis (pH < 5.9)[1–3]. Activation of TRPV1 by noxious stimuli induces inward cationic currents and the resulting action potentials in nociceptive DRG neurons then convey nociceptive information to the spinal dorsal horn[4]. Thus, TRPV1 is considered to be one of the major contributors of nociception[5].

Thermal hyperalgesia represents a pathological state in which the threshold of pain sensation to a thermal stimulus is decreased. Because TRPV1 has been strongly implicated in the thermal hyperalgesia developed in response to peripheral inflammation, it has been intensively investigated to help elucidate the mechanism underlying inflammatory thermal hyperalgesia[6,7]. It is well known that TRPV1 function is potentiated by inflammatory mediators, which can modify channel gating through receptors that activate the intracellular signaling cascades[6,8,9]. For instance, TRPV1 can be modified by protein kinases activated by pro-inflammatory factors, which can lower the activation threshold of the channel and thereby enhance its function[10,11]. Additionally, TRPV1 activity can be enhanced by inserting more channel proteins onto the plasma membrane through translocation from a

reserve in intracellular pools[12,13], as well as the transcriptional and translational control of TRPV1 expression[14].

While phosphorylation represents the best studied inflammation-related post-translational modification (PTM) mechanisms that exert functional modulation via either changes in plasma membrane expression or alterations in the biophysical properties of the TRPV1 channel, the effects of other types of PTM on this channel and their implications in inflammatory thermal hyperalgesia remain largely unexplored. As a form of PTM, SUMO modification has recently emerged as a key regulatory pathway of many biological processes[15,16]. SUMOylation modifies protein function by covalently binding a member of the SUMO family to the target protein. Such a modification can facilitate or prevent inter- and intra-molecular interactions via conformational changes or direct steric hindrance. An increasing number of ion channels, including GluK2-containing kainate receptors[17], voltage-gated potassium channels, Kv2.1[18] and Kv1.5[19], and a TRP channel, TRPM4[20], have been reported to be conjugated and regulated by SUMO, suggesting that SUMOylation may be a common mechanism in functional regulation of ion channels. The regulation by SUMO has been shown to control membrane trafficking, synaptic functions[15,17], and more recently nociception[21]. In the latest case, SUMOylation of a microtubule-binding protein, CRMP2, was shown to be required for proper

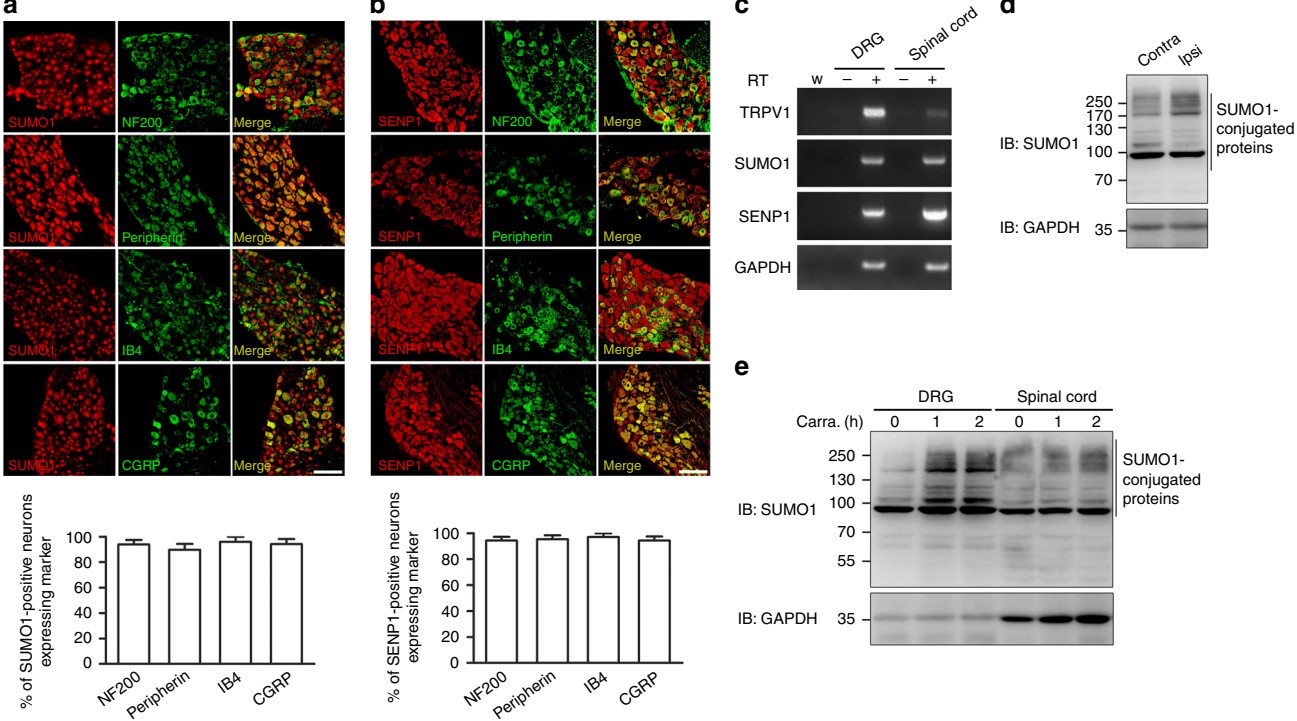

**Fig. 1** Expression of SUMO1 and SENP1 in DRG neurons. **a** Distribution of SUMO1 in DRG neurons. DRG neurons were permeabilized and double labeled with SUMO1 (red) and NF200, peripherin, IB4 and CGRP (green). The percents of SUMO1-positive neurons among those labeled with the indicated markers are shown in the bar graph below. The total numbers of neurons analyzed (*n*) ranged from 48 to 89 cells per condition. Data are means ± s.e.m, from three experiments. Scale bar: 100 μm. **b** Distribution of SENP1 in DRG neurons. Similar to **a** but SENP1 was used in place of SUMO1. The total numbers of neurons analyzed (*n*) ranged from 64 to 117 cells per condition. Data are means ± s.e.m, from three experiments. Scale bar: 100 μm. **c** RNA analysis by RT-PCR of TRPV1, SUMO1, SENP1, and GAPDH levels in DRG and spinal cord from adult wild-type mice. Negative controls: samples without reverse transcriptase (−RT) and water (W). Gel images are representatives of three independent experiments. **d** SUMO1-conjugated proteins were increased in DRG in the ipsilateral side of carrageenan-injected hindpaw as compared to the contralateral side. Carrageenan (2%, 20 μl) was administered by intraplantar injection into the left hindpaw of wild-type mice. L3–L4 DRG at the left (Ipsi) and right (Contra) sides were dissected 1 h after the injection and SUMO1-conjugated proteins were measured by western blotting. GAPDH was used as the loading control. Blots are representatives of three independent experiments. **e** SUMO1-conjugated proteins were increased in DRG but not spinal cord. At 0, 1, or 2 h after intraplantar injection of carrageenan (Carra.) into both hindpaws, L3–L4 DRG from both sides and spinal cord were dissected for western blotting for SUMO1 and GAPDH. Blots are representatives of three independent experiments

subcellular localization of the sodium channel subtype, $Na_V1.7$, and critically involved in neuropathic pain[21,22].

The SUMOylation state of a protein is determined by the balance between SUMOylation and deSUMOylation. Whereas SUMOylation is dynamically regulated by activity-dependent redistribution of SUMOylation machinery[16,23,24], it is rapidly reversed by the isopeptidase activity of SUMO/sentrin-specific proteases (SENPs), which strongly influences the conjugation/deconjugation balance of SUMO targeted proteins[25]. Here, we investigated the role of SUMOylation and deSUMOylation in inflammatory thermal hyperalgesia. We show that peripheral inflammation of mouse hindpaws by carrageenan injection enhances protein SUMOylation in DRG neurons, and conditional deletion of the sentrin-specific peptidase 1 (SENP1) gene in primary somatosensory neurons exacerbated thermal hyperalgesia in both carrageenan- and Complete Freund's adjuvant (CFA)-induced inflammation models. We further identified a lysine residue at the C-terminus of TRPV1 (K822) to be SUMOylated by SUMO1 and deSUMOylated by SENP1, which when mutated to Arg, not only failed to exhibit a SUMO1-induced channel sensitization to heat stimulus in vitro but also was unable to rescue inflammatory thermal hyperalgesia in vivo when introduced into DRG neurons of the *TRPV1* knockout mice.

## Results

### Peripheral inflammation enhances protein SUMOylation in DRG.
Sensory information from peripheral tissues is transmitted to the spinal cord via DRG neurons of small, medium, and large sizes, corresponding roughly to C-, Aδ-, and Aβ-fibers, respectively, with distinct functions[26]. Typically, the small-sized neurons transmit nociceptive information whereas the large ones are thought to mediate tactile and proprioceptive impulses[27]. To

assess a possible involvement of SUMOylation/deSUMOylation in sensory neurons, we first examined whether SUMO1 and SUMO1/SENP1 are expressed in mouse DRG. Immunohistochemical analysis revealed that both SUMO1 and SENP1 are expressed abundantly in small, medium, and large-sized DRG neurons, being present in nearly all A-fibers labeled by neurofilament NF200, peripherin-positive small-sized neurons, both non-peptidergic and peptidergic neurons labeled with isolectin B4 (IB4), and calcitonin gene-related peptide (CGRP), respectively (Fig. 1a, b). The presence of both SUMO1 and SENP1 mRNA in DRG was also detected by real-time polymerase chain reaction (RT-PCR) (Fig. 1c).

To test if inflammatory thermal hyperalgesia is accompanied with protein SUMOylation or deSUMOylation in the DRG, we employed a carrageenan paw edema model. One hour after intraplantar injection of carrageenan (2%, 20 µl) into the mouse left hindpaw, the level of SUMO1-conjugated proteins in lumbar 3/4 (L3–L4) DRG from the ipsilateral side was markedly increased as compared to that from the contralateral side (Fig. 1d). We then injected carrageenan into the mouse hindpaws at both sides and measured the levels of SUMO1-conjugated proteins in DRG and the spinal cord. The increase of protein SUMOylation in the DRG was maintained for at least 2 h, but no change was detected in the spinal cord (Fig. 1e), suggesting that changes in SUMO modification occur specifically in DRG neurons in response to peripheral inflammation.

### SENP1 cKO mice had greater inflammatory thermal hyperalgesia.
Because global deletion of SENP1 in mice causes anemia and embryonic lethality between E13.5 and postnatal day[28], we generated a conditional SENP1 knockout line (*SENP1flox/flox; Prrxl1-CreERT2*, *SENP1* cKO) by crossing a conditional

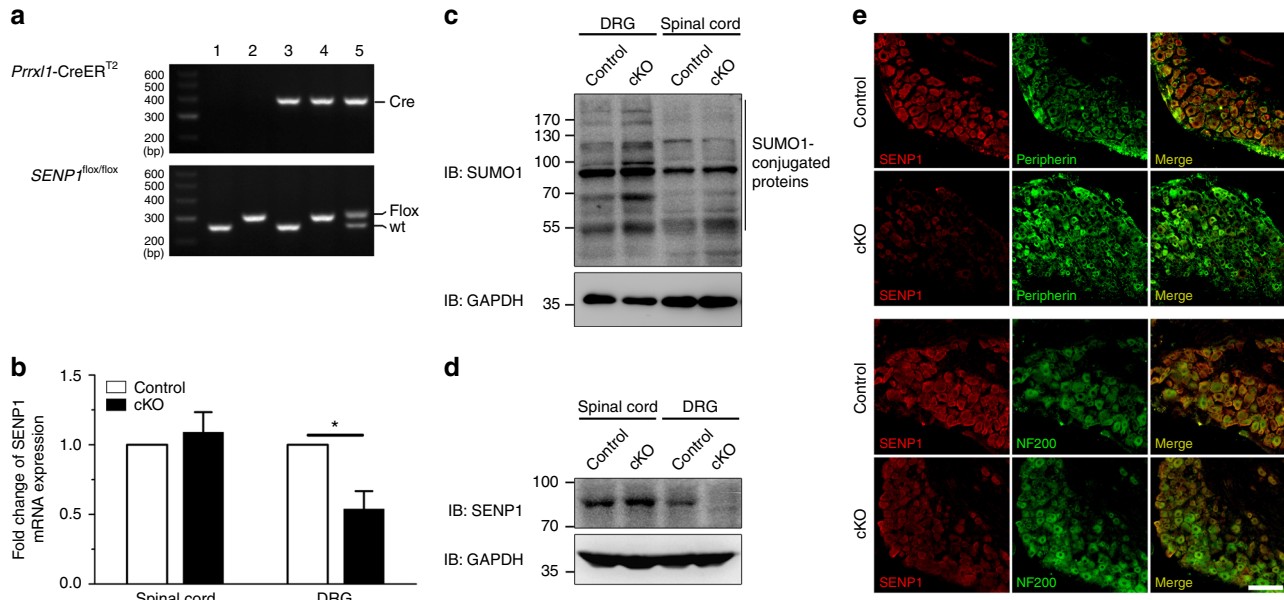

**Fig. 2** Generation of primary somatosensory neuron-specific *SENP1* cKO mice. **a** PCR analysis of genomic DNA obtained from DRG of *SENP1flox/flox* and *Prrxl1*-CreERT2 intercrosses. Line 1, wild type; line 2, *SENP1flox/flox* (Control); line 3, *Prrxl1*-Cre ERT2; line 4, *SENP1flox/flox*; *Prrxl1*-Cre ERT2 (*SENP1* cKO); line 5, *SENP1wt/flox*; *Prrxl1*-Cre ERT2. **b** SENP1 mRNA was reduced in DRG of *SENP1* cKO mice. Real-time qPCR was used to determine the SENP1 mRNA levels in the DRG and spinal cord from *SENP1* cKO and its littermate control mice 4 weeks after tamoxifen administration. Data are mean ± s.e.m. normalized to that of the control from three independent experiments. $P = 0.0227$, Control DRG vs. cKO DRG by Student's *t* test. **c** SUMO1-conjugated proteins were increased in DRG, but not spinal cord, of *SENP1* cKO mice. DRG and spinal cord were dissected from *SENP1* cKO and their littermate control mice and analyzed by western blotting using anti-SUMO1 and anti-GAPDH antibodies. **d** SENP1 protein level was decreased in *SENP1* cKO DRG compared to its littermate controls. DRG and spinal cord were dissected from *SENP1* cKO and their littermate control mice and analyzed by western blotting using anti-SENP1 and anti-GAPDH antibodies. **e** Verification of SENP1 gene deletion in DRG of *SENP1* cKO mice by immunohistochemistry, with double staining for SENP1 (red) and peripherin (green) or NF200 (green) for DRG sections from *SENP1* cKO mice (lower) and its littermate controls (upper). Scale bar: 100 µm

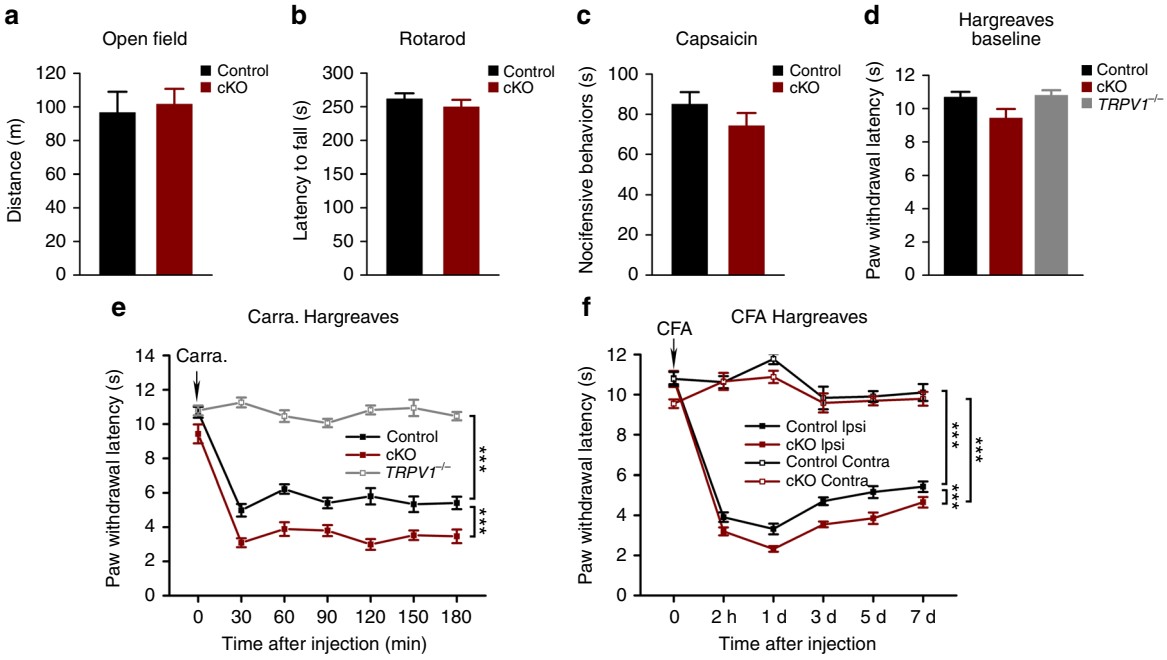

**Fig. 3** Increased inflammatory thermal hyperalgesia in *SENP1* cKO mice. **a** Locomotor activities and exploratory behaviors of uninjured 8-week-old *SENP1* cKO mice and its littermate controls were assessed by open field test. Total distances traveled during 30 min were: 96.6 ± 12.4 m for control (*n* = 14) and 101.6 ± 9.2 m for *SENP1* cKO (*n* = 6). **b** Motor coordination skills for *SENP1* cKO and their littermate control mice were assessed by rotarod test. The average latencies to fall from the rotating bar for three consecutive trials were: 261.7 ± 8.3 s for control (*n* = 7) and 249.7 ± 10.5 s for *SENP1* cKO mice (*n* = 10). **c** Nocifensive behaviors in response to intraplantar injection of capsaicin (2 μg) were tested for *SENP1* cKO and their littermate control mice. The time spent on licking, biting, and lifting (nocifensive behaviors) within the first 10 min of capsaicin injection were: 85.0 ± 6.0 s for control (*n* = 8) and 74.3 ± 6.4 s for *SENP1* cKO mice (*n* = 12). **d** Baseline withdrawal latencies of hindpaws to radiant heat were measured for *SENP1* cKO, its littermate controls and *TRPV1* $^{-/-}$ mice by Hargreaves's method. The paw withdrawal latencies (PWL) were: 10.51 ± 0.31 s for control (*n* = 15), 9.43 ± 0.55 s for *SENP1* cKO mice (*n* = 8), and 10.79 ± 0.31 s for *TRPV1* $^{-/-}$ (*n* = 5). **e** Time courses of PWL as assessed by the Hargreaves's method before (0 min) and at different time after intraplantar injection of carrageenan (2%, 20 μl) into the left hindpaws of *TRPV1* $^{-/-}$ mice, *SENP1* cKO mice and their littermate controls. *TRPV1* $^{-/-}$ mice did not develop inflammatory thermal hyperalgesia, whereas *SENP1* cKO mice exhibited markedly decreased PWL as compared to the control in the carrageenan edema model (Control, *n* = 15; *SENP1* cKO, *n* = 8; *TRPV1* $^{-/-}$, *n* = 5). ***P* < 0.0001, Control vs. *SENP1* cKO; ***P* < 0.0001, Control vs. *TRPV1* $^{-/-}$. **f** Time courses of PWL as assessed by the Hargreaves's method before (0 min) and at different times after intraplantar injection of CFA (0.5 mg ml$^{-1}$, 20 μl) into the left hindpaws of *SENP1* cKO mice and their littermate controls. Both the left (Ipsi) and right (Contra) paws were tested (Control, *n* = 12; *SENP1* cKO, *n* = 9). ***P* < 0.0001, Control Ipsi vs. *SENP1* cKO Ipsi; ***P* < 0.0001, Control Contra vs. Control Ipsi, and ***P* < 0.0001, *SENP1* cKO Contra vs. *SENP1* cKO Ipsi. Data are means ± s.e.m. Two-way ANOVA for **e** and **f**

*SENP1*$^{flox/flox}$ strain with a *Prrxl1* tamoxifen-inducible Cre line[29]. This resulted in a selective loss of SENP1 from DRG neurons (Fig. 2a). The mRNA and protein levels of SENP1 were markedly reduced in DRG of the *SENP1* cKO mice as determined by quantitative real-time PCR (qPCR), western blotting and immunohistochemical staining (Fig. 2b, d, e). Four weeks after tamoxifen administration, SENP1 mRNA in DRG of the cKO mice was reduced by >40% as compared to the littermate controls (Fig. 2b), while no change was detected in the spinal cord where *Prrxl1* tamoxifen-inducible-Cre is weakly expressed (Fig. 2b). As expected from the loss of a deSUMOylating enzyme, the level of SUMO1-conjugated proteins was also markedly increased in DRG but not in the spinal cord of the 8-week-old *SENP1* cKO mice as compared to the littermate controls (Fig. 2c).

To evaluate the effect of SENP1 knockdown in sensory neurons on neurological functions, the *SENP1* cKO mice and their littermate controls were subjected to open field, rotarod and Hargreaves tests. No significant difference was detected between genotypes in the open field and rotarod tests, indicating that there are no gross motor abnormalities (Fig. 3a, b). Intraplantar injection of a low dose of capsaicin (0.2 μg, 10 μl), which induces nocifensive behaviors without evoking neurogenic inflammation and thermal hyperalgesia as in the case of high doses[30], also resulted in similar responses between *SENP1* cKO mice and the

littermate controls in terms of time spent in lifting, licking, and biting the injected site (Fig. 3c). Furthermore, the baseline paw withdrawal latencies evoked by radiant heat were similar between the two groups, averaging at 10.69 ± 0.32 s for control and 9.43 ± 0.55 s for the *SENP1* cKO mice (Fig. 3d). Therefore, the basal neurological function and sensations to heat and capsaicin were not changed by the selective knockout of SENP1 expression from DRG neurons.

We then tested the effect of peripheral inflammation on thermosensation in *SENP1* cKO mice and littermate controls. After intraplantar injection of carrageenan into the left hindpaws of control mice, the withdrawal latency of the injected paws in response to radiant heat decreased dramatically to about 47% of the baseline level within 30 min and the effect lasted for at least 3 h (Fig. 3e). Consistent with previous studies[3,5], this effect did not occur in TRPV1 knockout (*TRPV1* $^{-/-}$) mice (Fig. 3e), showing that the carrageenan-induced thermal hyperalgesia is dependent on TRPV1. Remarkably, the carrageenan injection resulted in a stronger reduction of the paw withdrawal latency in the *SENP1* cKO mice than their littermate controls, reaching the lowest value of 2.99 ± 0.31 s at 2 h after carrageenan injection, which remained lower than control throughout the 3-h test period (Fig. 3e). These results suggest that SENP1 deficiency in DRG neurons aggravates thermal hyperalgesia in response to peripheral inflammation

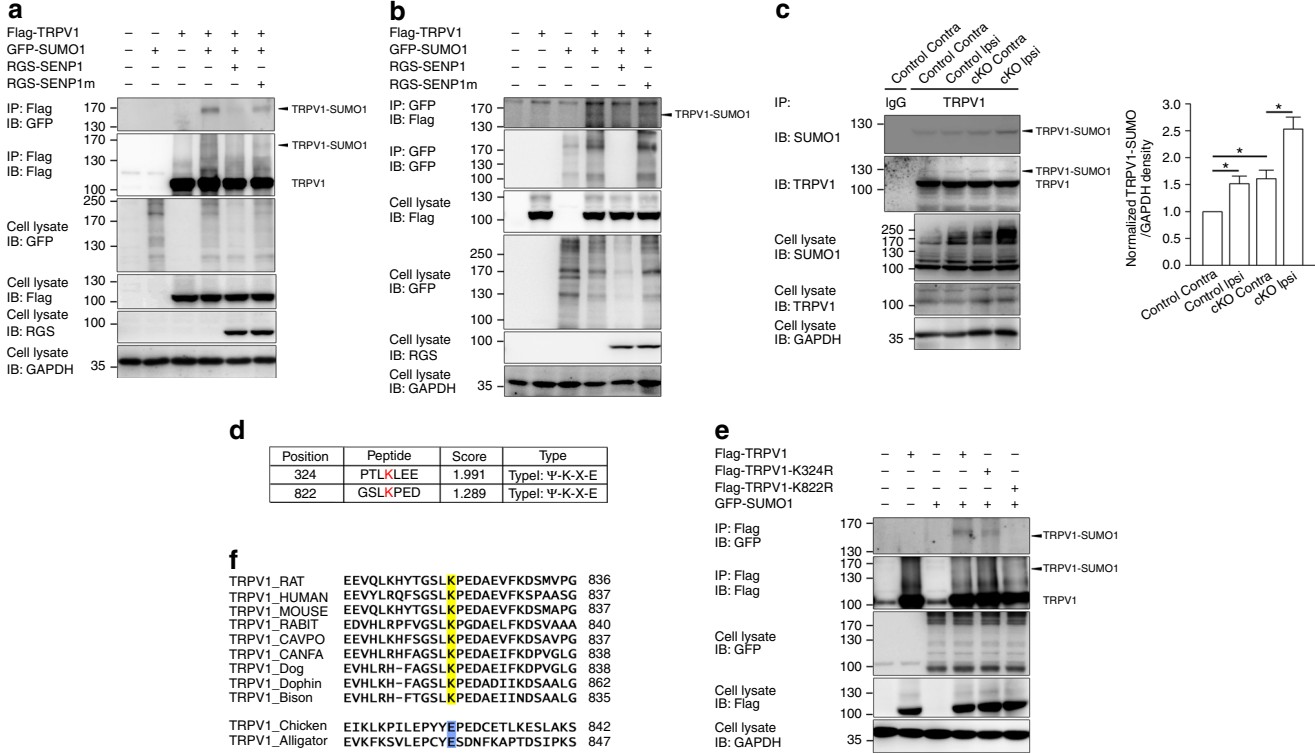

**Fig. 4** SUMO1 modification of TRPV1. **a** Conjugation of TRPV1 by GFP-SUMO1 co-expressed in CHO-K1 cells. Cells were transiently co-transfected with the following plasmids as indicated: Flag-TRPV1, GFP-SUMO1, RGS-SENP1, and RGS-SENP1m. Cell lysates were immunoprecipitated (IP) with anti-Flag antibody and analyzed by immunoblotting (IB) using anti-GFP or anti-Flag. Whole-cell lysates were also used for IB with anti-GFP, anti-Flag, and anti-RGS antibodies for input. Molecular weight standards (in kD) are shown on the left. **b** Similar to **a** for reciprocal IP using anti-GFP (for GFP-SUMO1) and IB by anti-Flag (for Flag-TRPV1). **c** Increased TRPV1 SUMOylation in DRG from the ipsilateral side of carrageenan-injected hindpaw of *SENP1* cKO mice. Carrageenan (2%, 20 μl) was injected into the left hindpaw of *SENP1* cKO and littermate control mice. L3–L4 DRG were dissected from the ipsilateral (Ipsi) and contralateral (Contra) sides 1 h after the injection. Lysates of DRG isolated from littermate control (Control) and *SENP1* cKO (cKO) mice were subjected to IP with a control IgG or an anti-TRPV1 antibody, which were followed by IB using anti-SUMO1 and anti-TRPV1 antibodies. Note the higher level of SUMOylated TRPV1 in the Ipsi than the Contra side and the higher levels in *SENP1* cKO than Control. Quantification of western blots in **c** were determined by measuring the relative intensity of TRPV1-SUMO band and the corresponding GAPDH band from the same lysate, which was then compared with Control performed in parallel in the same experiment. Data shown are means ± s.e.m. for three experiments. $P = 0.0211$, Control Contra vs. Control Ipsi; $P = 0.0180$, Control Contra vs. cKO Contra; and $P = 0.0274$ cKO Contra vs. cKO Ipsi by Student's *t* test. **d** Potential SUMO modification sites in rat TRPV1 predicted by SUMOsp 2.0 software. **e** Loss of TRPV1 SUMOylation by K822R mutation. K324R and K822R substitutions in Flag-TRPV1 were made by site-directed mutagenesis and co-expressed with GFP-SUMO1 as indicated. Cell lysates were subjected to IP by anti-Flag, followed by IB for GFP and Flag. **f** Alignment of TRPV1 sequences from different species as indicated at the consensus SUMOylation site, with the conserved lysine in mammalian species shown in yellow. Note in chicken and alligator, the lysine is replaced by glutamate shown in blue. All blot images are representatives of three independent experiments

induced by carrageenan. To test if this effect is generally applicable to other inflammation models, we also used CFA to induce edema in mouse hindpaws. The CFA model is thought to mimic both acute and chronic phases of inflammatory pain[31]. After intraplantar injection of CFA into the left hindpaws of control mice, the withdrawal latency of the injected paws to radiant heat decreased to about 36% of the baseline level within 2 h and the effect lasted for at least 7 days (Fig. 3f). Again, the decrease was more dramatic in the *SENP1* cKO mice, reaching 3.20 ± 0.20 s and 2.32 ± 0.14 s in 2 and 24 h, respectively, and the hypersensitivity continued for 7 days (Fig. 3f). Therefore, the reduced ability to deSUMOylate because of the SENP1 deficiency in DRG neurons exacerbated inflammatory thermal hyperalgesia in both carrageenan and CFA-induced edema models.

**SUMO conjugates TRPV1 at Lys-822.** Because TRPV1 is necessary for the development of inflammatory thermal hyperalgesia (Fig. 3e), SUMO modification of TRPV1 or an associated protein(s) either upstream or downstream of TRPV1 signaling

may be involved in this process such that the deficiency of SENP1 rendered the mice more sensitive or hyperalgesic to heat in response to peripheral inflammation.

To test if TRPV1 can be SUMOylated and the functional consequence of such a modification, we first examined whether Flag-tagged rat TRPV1 expressed in CHO-K1 cells could be labeled by the overexpressed GFP-SUMO1. Because TRPV1 expressed in heterologous systems is glycosylated, yielding bands in the range of ~125–135 kD (Supplementary Fig. 1a), which interferes with the detection of SUMOylated TRPV1 (estimated to be ~120 kD), we used GFP-SUMO1 instead of SUMO1. This gives an expected size of ~160 kD, away from the glycosylated TRPV1. Indeed, the GFP antibody detected a protein band of ~160 kD in the precipitants pulled down by the anti-Flag antibody via co-immunoprecipitation (co-IP) under denaturing conditions (Fig. 4a) and reciprocally, the Flag-tagged TRPV1 was also pulled down by a GFP antibody (Fig. 4b). The SUMOylation of TRPV1 was abolished by the co-expression of RGS-tagged SENP1 but not of a catalytically dead SENP1 mutant (SENP1m) (Fig. 4a, b), confirming that SENP1 is involved in TRPV1

deSUMOylation. The treatment with PNGase F removed the glycosylated bands at ~125–135 kD, but did not alter the 160 kD band (Supplementary Fig. 1b, d). In cells that co-expressed GFP-SUMO1 and the non-glycosylated Flag-tagged TRPV1 mutant (Asn604 changed to Thr, N604T)[32,33], the 160-kD GFP-positive band was also pulled down by the Flag antibody (Supplementary Fig. 1c, d). These not only confirm that the 160 kD band represents GFP-SUMO1-labeled TRPV1, but also indicate that glycosylation was not required for TRPV1 SUMOylation. In addition, TRPV1 SUMOylation was confirmed using mouse TRPV1 expressed in CHO-K1 cells (Supplementary Fig. 2a) and rat TRPV1 modification by His-tagged SUMO1 in HEK293T cells (Supplementary Fig. 2b).

As a control, we tested the ability of SUMO1 to co-IP with another sensory neuron channel, acid sensing ion channel 3 (ASIC3), which is also known to participate in inflammatory pain[34]. Accumulating evidence has shown that while TRPV1 is involved in inflammatory thermal hyperalgesia, ASIC3 participates only in mechanical but not thermal hyperalgesia in response to carrageenan-induced paw inflammation. In addition, ASIC3 is also involved in chronic pain induced by repeated acid injection[35,36]. Interestingly, upon co-expression of GFP:HA-ASIC3 and Flag-SUMO1 in CHO-K1 cells, the anti-Flag antibody failed to pull down any SUMO1-positive band in the size range that would be expected for SUMOylated GFP:HA-ASIC3 (Supplementary Fig. 3). Therefore, ASIC3 could not be SUMOylated under these experimental conditions and the above results demonstrate that between the two channels involved in inflammatory pain, only TRPV1 can be SUMOylated by SUMO1 and deSUMOylated by SENP1.

Next, we performed co-IP using lysates prepared from mouse DRG under denaturing conditions. Because glycosylation of TRPV1 is very weak in freshly isolated DRG without peripheral processes, which contain mainly neuronal cell bodies[37,38], a SUMO1-positive band of ~125 kD was readily detectable in precipitants pulled down by the anti-TRPV1 antibody (Fig. 4c). The amount of SUMOylated proteins pulled down by the anti-TRPV1 antibody was higher in DRG lysates obtained from the *SENP1* cKO mice than their littermate controls (Fig. 4c), and was increased in DRG from the ipsilateral as compared to that from the contralateral side of animals that received intraplantar injection of carrageenan 1 h prior to the tissue dissection (Fig. 4c). These results demonstrate that native TRPV1 in DRG neurons undergoes SUMOylation/deSUMOlylation under physiological conditions and the level of SUMOylation is downregulated by

SENP1. However, upon peripheral inflammation, TRPV1 SUMOylation is enhanced in the DRG neurons. This likely contributes to the development of inflammatory thermal hyperalgesia, as it was exacerbated by *SENP1* cKO mice from the sensory neurons (Fig. 3e, f).

Protein SUMOylation typically occurs at lysine residues located within the consensus sequence Ψ–K–X–E, where Ψ is any hydrophobic residue and X is any residue. Analysis of rat TRPV1 protein sequence revealed two lysine (Lys) residues at positions 324 and 822 that adhere to the consensus (Fig. 4d). We therefore mutated these residues to arginine (R) individually and tested the ability of GFP-SUMO1 to conjugate them after co-expression in CHO-K1 cells. While K324R became SUMOylated similarly as the wild-type TRPV1, K822R were not SUMOylated (Fig. 4e), indicating that K822 is the primary residue of TRPV1 subjected to SUMO1 modification. Moreover, the K822R mutation abolished SUMOylation without affecting glycosylation (Supplementary Fig. 1e), showing that the two forms of PTM are independent of each other. It is also intriguing that K822 is only conserved among mammals (e.g., in rat, human, mouse, rabbit, guinea pig, and dog, but not chicken and alligator) (Fig. 4f), indicating that SUMOylation of TRPV1 at the C-terminal Lys residue is mammalian specific. Taken together, these findings indicate that TRPV1 is SUMOylated at the C-terminal residue Lys-822, which is downregulated by SENP1.

**SUMOylation alters thermal sensitivity of TRPV1.** TRPV1 is gated by a variety of stimuli including temperature, protons, and several 'vanilloid-like' endogenous ligands[39]. Any or all of these could be affected by SUMOylation. A number of mechanisms have been suggested to underlie inflammatory thermal hyperalgesia, including a drop of the temperature threshold ($T_{threshold}$) of TRPV1 activation to below the normal physiological temperature, upregulation of TRPV1 protein expression, recruitment of more TRPV1-positive sensory afferents, and modifications of biophysical properties of the TRPV1 channel[40,41]. To learn how SUMOylation affects TRPV1 function, we first measured ionic currents elicited by the TRPV1 agonist, capsaicin, using whole-cell patch clamp recording. With the cells held at −60 mV, capsaicin (1 μM) evoked similar current densities between CHO-K1 cells that expressed TRPV1 and TRPV1-K822R (Fig. 5a and Fig. 6a, c). The amounts of surface expressed and total Flag-TRPV1 proteins were determined by surface biotinylation and western blotting, which revealed comparable levels of total and

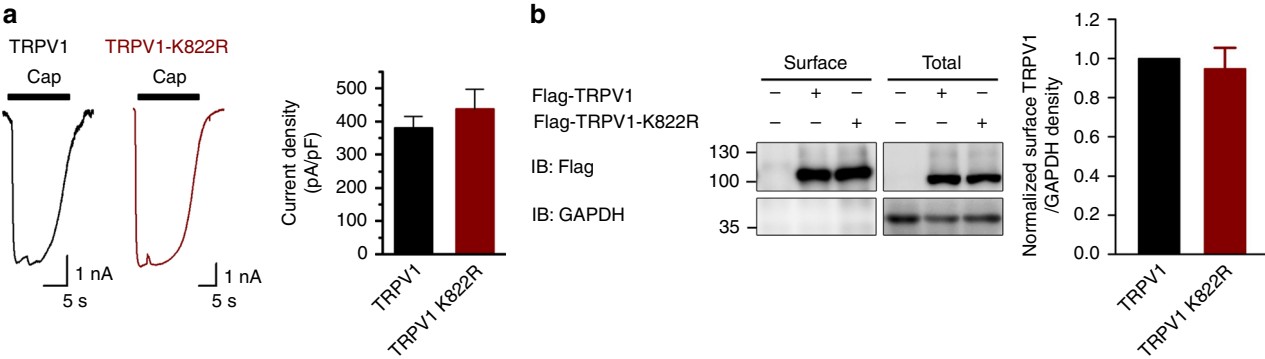

**Fig. 5** K822R mutation of TRPV1 did not alter its membrane surface expression and capsaicin-evoked whole-cell current density. **a** CHO-K1 cells were transfected with Flag-TRPV1 or Flag-TRPV1-K822R. Representative traces (left) and summary of current density (right) for capsaicin (1 μM)-activated whole-cell currents at −60 mV are shown. The current densities are: TRPV1, 345 ± 56 pA/pF ($n$ = 7) and TRPV1-K822R, 398 ± 59 pA/pF ($n$ = 5). Data are means ± s.e.m. **b** CHO-K1 cells were transfected with Flag-TRPV1 or Flag-TRPV1-K822R. Surface levels of TRPV1 were measured by IB after plasma membrane proteins were biotinylated and purified with streptavidin-agarose. GAPDH were used as controls for cytoplasmic proteins, respectively. Blots are representatives of at least three independent experiments

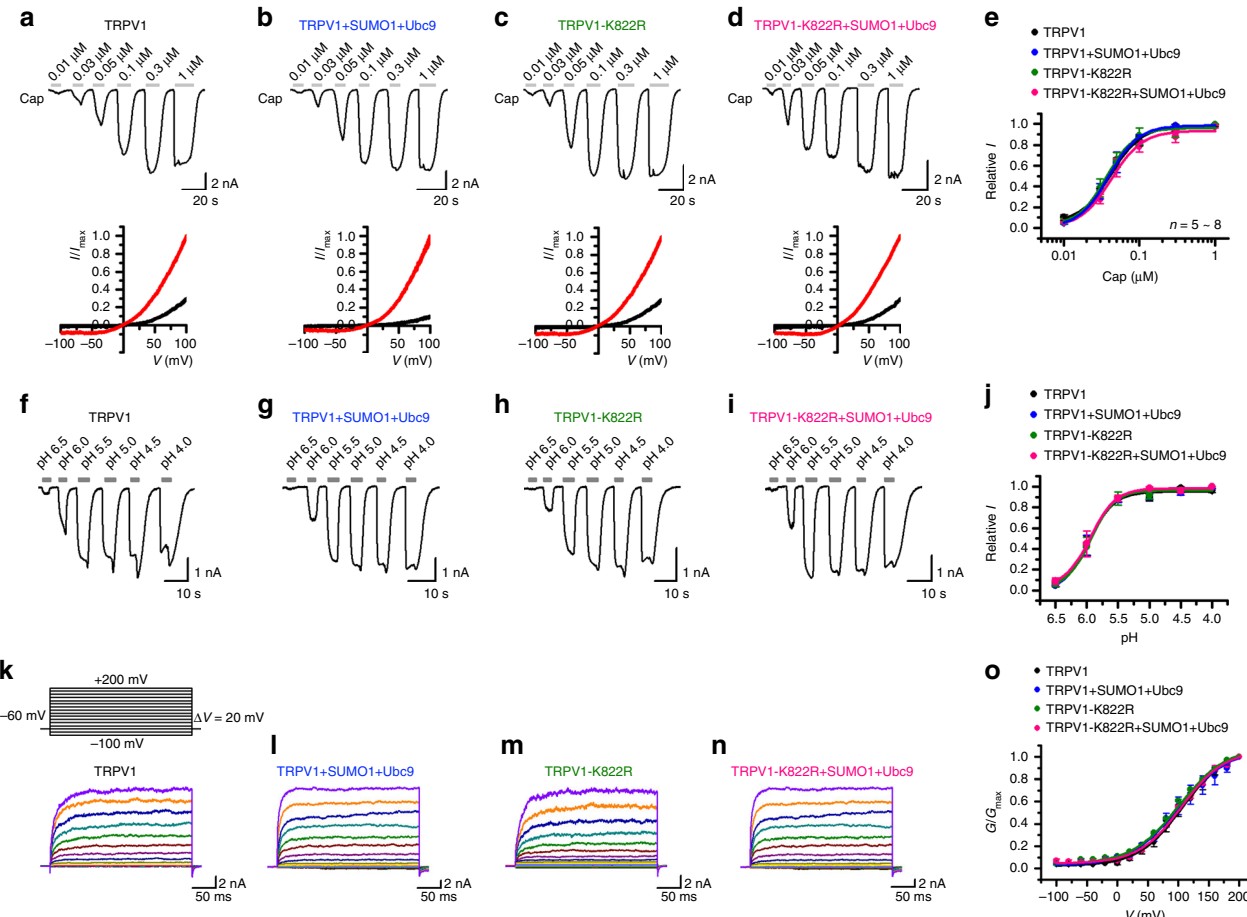

**Fig. 6** SUMOylation had no effect on capsaicin-, proton- and voltage-dependent activation of TRPV1 channels. **a**, **b** Representative traces for currents at −60 mV evoked by increasing concentrations of capsaicin (Cap) as indicated for CHO-K1 cells that expressed Flag-tagged rat TRPV1 (Flag-TRPV1) (**a**), Flag-TRPV1+GFP-SUMO1+Ubc9 (**b**), Flag-TRPV1-K822R (**c**), and Flag-TRPV1-K822R+GFP-SUMO1+Ubc9 (**d**). The current–voltage relationships shown below the traces are for basal (black) and 0.03 μM capsaicin-stimulated (red) conditions. **e** Concentration–response curves for capsaicin-evoked currents. Solid lines indicate fits with the Hill equation. Data points are means ± s.e.m., with $n = 8$ for TRPV1 (black); $n = 6$ for TRPV1+SUMO1+Ubc9 (blue); $n = 6$ for TRPV1-K822R (olive); $n = 6$ for TRPV1-K822R+SUMO1+Ubc9 (pink). **f–i** Representative traces for currents at −60 mV evoked by solutions with decreasing pH values as indicated for CHO-K1 cells that expressed Flag-TRPV1 (**f**), Flag-TRPV1+GFP-SUMO1+Ubc9 (**g**), Flag-TRPV1-K822R (**h**), and Flag-TRPV1-K822R+GFP-SUMO1+Ubc9 (**i**). **j** Concentration–response curves for proton-evoked currents. Solid lines indicate fits with the Hill equation. Data points are means ± s.e.m., with $n = 7$ for TRPV1 (black); $n = 9$ for TRPV1+SUMO1+Ubc9 (blue); $n = 7$ for TRPV1-K822R (olive); $n = 7$ for TRPV1 K822R +SUMO1+Ubc9 (pink). **k–n** Representative traces of currents evoked by a family of voltage steps from −100 to +200 mV with 20 mV increments as shown in inset in **k** for CHO-K1 cells that expressed Flag-TRPV1 (**k**), Flag-TRPV1+GFP-SUMO1+Ubc9 (**l**), Flag-TRPV1-K822R (**m**), and Flag-TRPV1-K822R +GFP-SUMO1+Ubc9 (**n**). Holding potential was −60 mV. **o** Conductance–voltage (G–V) relationships derived from the experiments shown in **k–m** fitted with the Boltzmann function, which yielded the following results: for TRPV1 (black), $V_{1/2} = 105.8 \pm 2.2$ mV, and $\kappa = 35.5 \pm 1.9$ ($n = 7$); TRPV1+SUMO1 +Ubc9 (blue), $V_{1/2} = 101.5 \pm 4.2$ mV, and $\kappa = 40.2 \pm 3.7$ ($n = 7$); TRPV1-K822R (olive), $V_{1/2} = 101.3 \pm 3.7$ mV, and $\kappa = 38.9 \pm 1.7$ ($n = 7$); and TRPV1-K822R +SUMO1+Ubc9 (pink), $V_{1/2} = 104.8 \pm 1.5$ mV, and $\kappa = 35.1 \pm 1.3$ ($n = 5$). Data points are means ± s.e.m

surface expression between wild-type TRPV1 and TRPV1-K822R (Fig. 5b). To examine the concentration dependence, the individual cells were exposed to increasing concentrations of capsaicin, with a thorough washout between the two consecutive applications (Fig. 6a, c). Fitting the current amplitudes normalized to that evoked by 1 μM capsaicin with the Hill equation yielded similar $EC_{50}$ values and Hill coefficients ($n_H$) for TRPV1 and TRPV1-K822R (Fig. 6e, $EC_{50} = 42.5 \pm 5.4$ nM, $n_H = 2.1 \pm 0.6$ for TRPV1 and $EC_{50} = 36.5 \pm 3.0$ nM, $n_H = 2.4 \pm 0.4$ for TRPV1-K822R). Moreover, the co-expression of SUMO1 and HA-Ubc9 with TRPV1 or TRPV1-K822R did not alter the maximal response and concentration dependence to capsaicin (Fig. 6b, d, e, $EC_{50} = 38.8 \pm 0.9$ nM, $n_H = 2.2 \pm 0.1$ for TRPV1+SUMO1+Ubc9 and $EC_{50} = 42.5 \pm 5.4$ nM, $n_H = 2.1 \pm 0.6$ for TRPV1-K822R +SUMO1+Ubc9). Therefore, the SUMO1 modification of TRPV1 did not appear to alter capsaicin-evoked currents.

To examine if SUMOylation could affect proton activation of TRPV1 currents, we applied variable acidic solutions covering a broad pH range from 6.5 to 4.0 to individual cells and determined $pH_{0.5}$ and $n_H$ (Fig. 6f–j). However, no significant difference was noticed between TRPV1 and TRPV1-K822R with or without the co-expression of SUMO1+Ubc9 ($pH_{0.5} = 6.0 \pm 0.1$, $n_H = 2.1 \pm 0.6$ for TRPV1; $pH_{0.5} = 6.0 \pm 0.1$, $n_H = 2.2 \pm 0.4$ for TRPV1+SUMO1 +Ubc9; $pH_{0.5} = 5.9 \pm 0.2$, $n_H = 2.3 \pm 0.6$ for TRPV1-K822R; $pH_{0.5} = 5.9 \pm 0.1$, $n_H = 2.2 \pm 0.3$ for TRPV1-K822R+SUMO1 +Ubc9). Therefore, the SUMOylation did not alter the sensitivity of TRPV1 to acid either.

Next, we examined the voltage dependence of TRPV1 and TRPV1-K822R in the absence and presence of co-expressed SUMO1+Ubc9 using a voltage step protocol. From the holding potential of −60 mV, 200 ms depolarizing step pulses from −100 to 200 mV in 20 mV increments were applied to individual cells

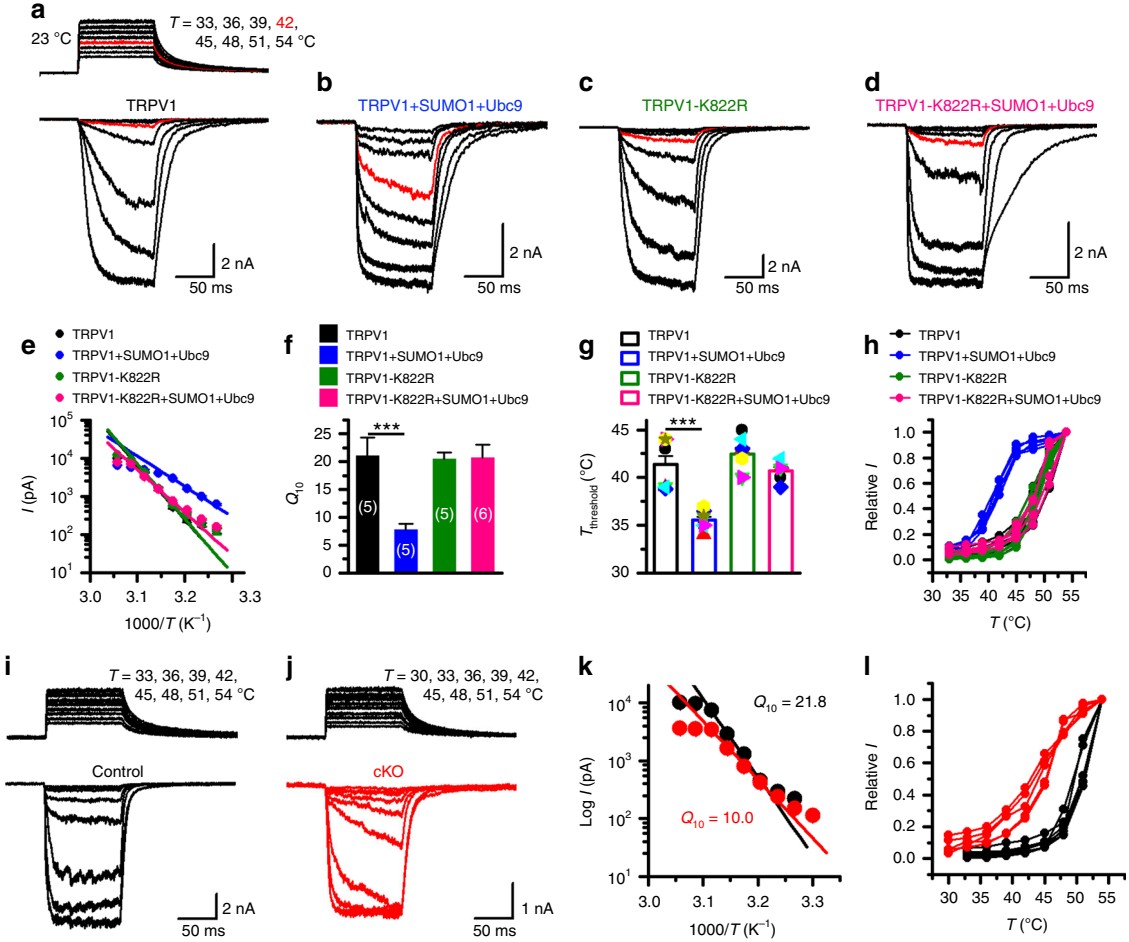

**Fig. 7** SUMOylation lowered temperature coefficient ($Q_{10}$) and temperature threshold ($T_{threshold}$) of TRPV1 activation. **a–d** Representative responses to a family of temperature jumps ranging from 33 to 54 °C (see inset in **a** for an example) for CHO-K1 cells that expressed Flag-TRPV1 (**a**), Flag-TRPV1+GFP-SUMO1+Ubc9 (**b**), Flag-TRPV1-K822R (**c**), and Flag-TRPV1-K822R+GFP-SUMO1+Ubc9 (**d**). Temperature was calibrated offline from the pipette current using the temperature dependence of electrolyte conductivity. The red traces indicate the response at 42 °C. **e** Arrhenius plot of steady-state currents shown in **a–d**. The major component of the reflection that represents the strong temperature dependence was fitted to a linear equation. Error bars indicate standard deviations (s.d.). **f, g** Comparison of $Q_{10}$ derived from the linear fits in **e** and $T_{threshold}$ changes. $Q_{10} = 21.0 \pm 3.3$ ($n = 5$) and $T_{threshold} = 41.4 \pm 0.9$ ($n = 8$) for TRPV1, $Q_{10} = 7.8 \pm 1.0$ ($n = 5$) and $T_{threshold} = 35.6 \pm 2.6$ ($n = 8$) for TRPV1+SUMO1+Ubc9, $Q_{10} = 20.5 \pm 1.2$ ($n = 5$) and $T_{threshold} = 42.4 \pm 0.7$ ($n = 7$) for TRPV1-K822R, $Q_{10} = 20.7 \pm 2.3$ ($n = 6$) and $T_{threshold} = 40.7 \pm 0.4$ ($n = 6$) for TRPV1-K822R+SUMO1+Ubc9. Colored symbols in **g** indicate individual data points. Data are means ± s.e.m. $P = 0.0004$ for $Q_{10}$ TRPV1 vs. TRPV1+SUMO1+Ubc9; $P = 0.00004$ for $T_{threshold}$ TRPV1 vs. TRPV1+SUMO1 +Ubc9 by Student's $t$ test. **h** Temperature response curves for TRPV1 (black), TRPV1+SUMO1+Ubc9 (blue), TRPV1-K822R (olive), and TRPV1-K822R +SUMO1+Ubc9 (pink), measured from the maximal currents at the end of temperature steps. Each curve represents measurements from an individual cell and the responses were normalized to its maximum responses at 54 °C. **i, j** Representative whole-cell current traces evoked by a family of temperature jumps in DRG neurons isolated from control and *SENP1* cKO mice. Holding potential was −60 mV. **k** Arrhenius plots for cells shown in **i, j**. Heat-evoked responses were fitted by linear regression and yielded $Q_{10} = 21.8$ for control (black, $n = 6$) and $Q_{10} = 10.0$ for *SENP1* cKO (red, $n = 5$). **l** Comparison of temperature response curves of control (black) and *SENP1* cKO (red) DRG neurons

(Fig. 6k, inset), which evoked non-inactivating ionic currents at voltages >0 mV for all cells (Fig. 6k–n). The current at the end of the 200 ms pulse was converted to conductance and normalized to that obtained at 200 mV before fitting with the Boltzmann function, which yielded similar conductance–voltage ($G$–$V$) relationships for both TRPV1 constructs no matter if SUMO1 was present or not (Fig. 6o). These indicate that neither the mutation nor co-expression of SUMO1 changed the voltage dependence of TRPV1 under basal conditions at the room temperature.

Last, we tested if TRPV1 SUMOylation could affect the temperature response of the channel. Although temperature sensing of TRPV1 may be tightly linked to voltage-dependent gating[42], the large and quick changes in enthalpy and entropy within milliseconds during heat activation of the channel suggest

that the temperature gating of TRPV1 is accomplished by a rapid and substantial structural shift from the closed to the open state and is best studied by applying quick heat steps to allow fast and concerted conformational transitions[43]. To do so, we used an infrared laser diode system to locally increase the temperature surrounding the cell being recorded at millisecond speed[43]. CHO-K1 cells expressing TRPV1 were held at −60 mV when the temperature jumps were delivered (Fig. 7a, inset). Interestingly, although the co-expression of SUMO1+Ubc9 did not alter TRPV1 currents evoked by high temperatures (51 and 54 °C), it clearly resulted in an elevated response to temperatures near the threshold (~42–43 °C) of TRPV1 activation under physiological conditions[1] (Fig. 7a, b). Using Arrhenius plot (Fig. 7e), we determined the temperature coefficient ($Q_{10}$) of TRPV1 to be decreased by ~60% (Fig. 7f) and the $T_{threshold}$ dropped by ~6 °C

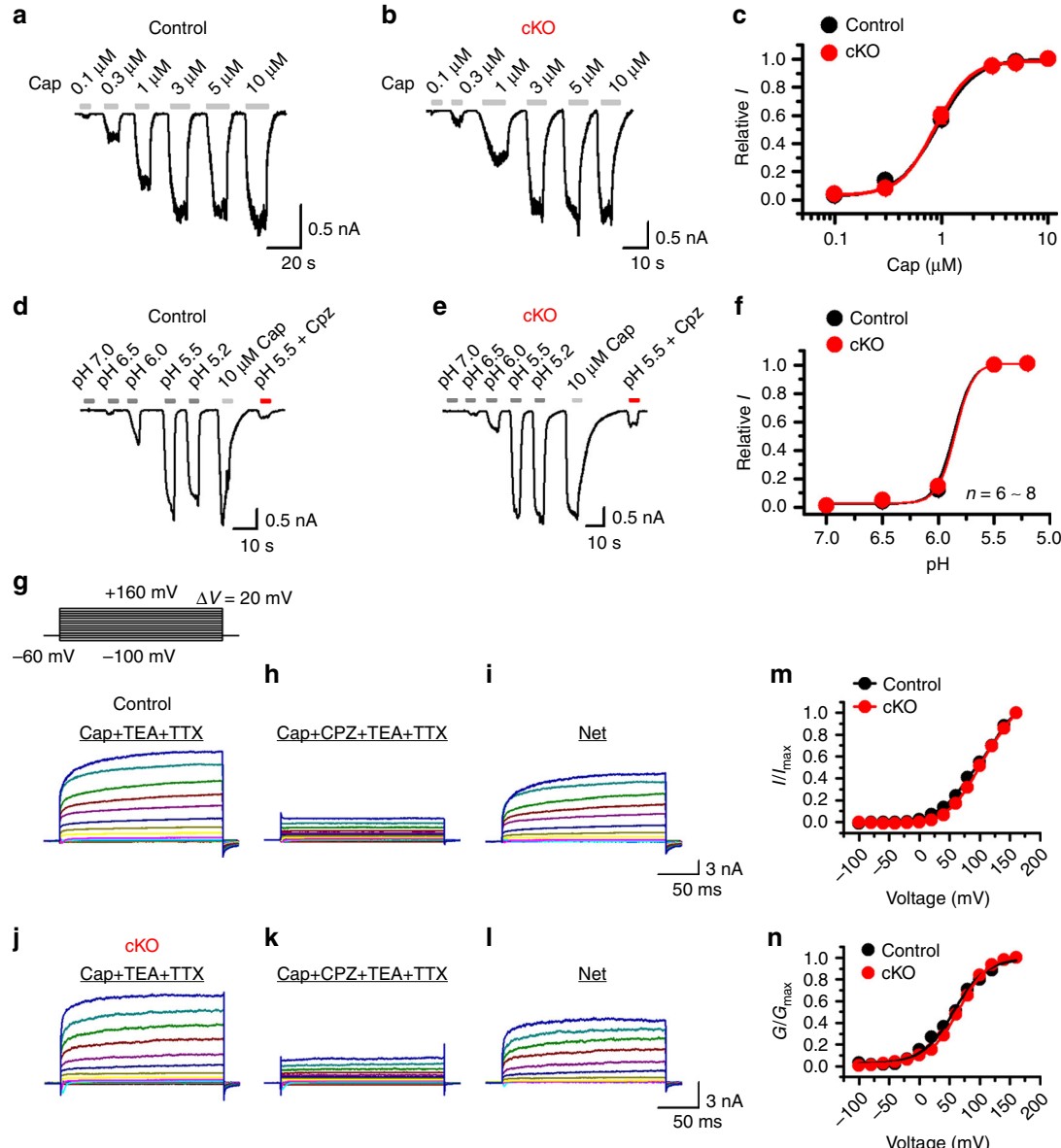

**Fig. 8** SUMOylation had no effect on capsaicin- and proton-evoked and the voltage dependence of TRPV1 channel activation in DRG neurons. **a**, **b** Representative whole-cell currents at −60 mV evoked by varying concentrations of capsaicin (Cap) as indicated in DRG neurons isolated from control (**a**) and *SENP1* cKO mice, respectively. Holding potentials = −60 mV. **c** Concentration responses to capsaicin for control and *SENP1* cKO DRG neurons. Data were fitted with Hill equation, which yielded: control, $EC_{50} = 0.91 \pm 0.01$ μM, and $n_H = 2.40 \pm 0.15$ ($n = 8$); *SENP1* cKO, $EC_{50} = 0.87 \pm 0.02$ μM and $n_H = 2.68 \pm 0.28$ ($n = 10$). **d**, **e** Representative whole-cell currents at −60 mV in responses to bath solutions of varying pH as indicated in control and *SENP1* cKO DRG neurons, respectively. The cell was also stimulated with 10 μM Cap and then pH 5.5 combined with 10 μM capsazepine (Cpz) to confirm that the currents were mediated by TRPV1. **f** Dose responses to pH for control and *SENP1* cKO TRPV1-positive DRG neurons. Data were fitted with Hill equation, which yielded: control, $pH_{0.5} = 5.9 \pm 0.1$, $n_H = 5.3 \pm 2.4$ ($n = 8$); *SENP1* cKO, $pH_{0.5} = 5.8 \pm 0.1$, $n_H = 5.6 \pm 3.2$ ($n = 7$). **g–l** Representative whole-cell currents elicited by the voltage step protocol shown in the inset in **g** for control (**g–i**) and *SENP1* cKO (**j–l**) DRG neurons. The cell was exposed to 0.3 μM Cap combined with 10 mM TEA-Cl and 1 μM TTX (**g** and **j**) or 0.3 μM Cap combined with 10 mM TEA-Cl, 1 μM TTX, and 10 μM Cpz (**h** and **k**), in order to isolate the net TRPV1-mediated currents through subtraction (**i** and **l**). *I–V* (**m**) and *G–V* (**n**) relationships of TRPV1 for control and *SENP1* cKO DRG neurons. Data in **n** were fitted with Boltzmann equation, which yielded: control $V_{1/2} = 56.9 \pm 2.6$ mV, and slope factor = $26.2 \pm 2.3$; *SENP1* cKO, $V_{1/2} = 65.4 \pm 1.1$ mV, and slope factor = $25.2 \pm 1.0$

(Fig. 7g) in the presence of SUMO1+Ubc9. Plotting the relative responses against the step temperatures also clearly revealed a left-shift of the temperature dependence of TRPV1 by SUMO1 +Ubc9 (Fig. 7h). However, the co-expression of SUMO1+Ubc9 did not affect the $Q_{10}$ and $T_{threshold}$ of TRPV1-K822R (Fig. 7c–h), indicating that SUMO1 acted by modifying K822 to alter the temperature sensitivity of TRPV1. The specific modulation by SUMO1 on temperature sensing was also observed using CHO-

K1 cells that expressed mouse TRPV1 and its K823R mutant (Supplementary Fig. 4), as well as using HEK293T cells that expressed rat (Supplementary Fig. 5) and mouse TRPV1 (Supplementary Fig. 6). It was also readily detectable without co-expressing Ubc9 with SUMO1 (Supplementary Fig. 7).

To evaluate if SUMOylation is also important for temperature sensing of native TRPV1 channels, we examined temperature-evoked currents in DRG neurons isolated from wild-type and

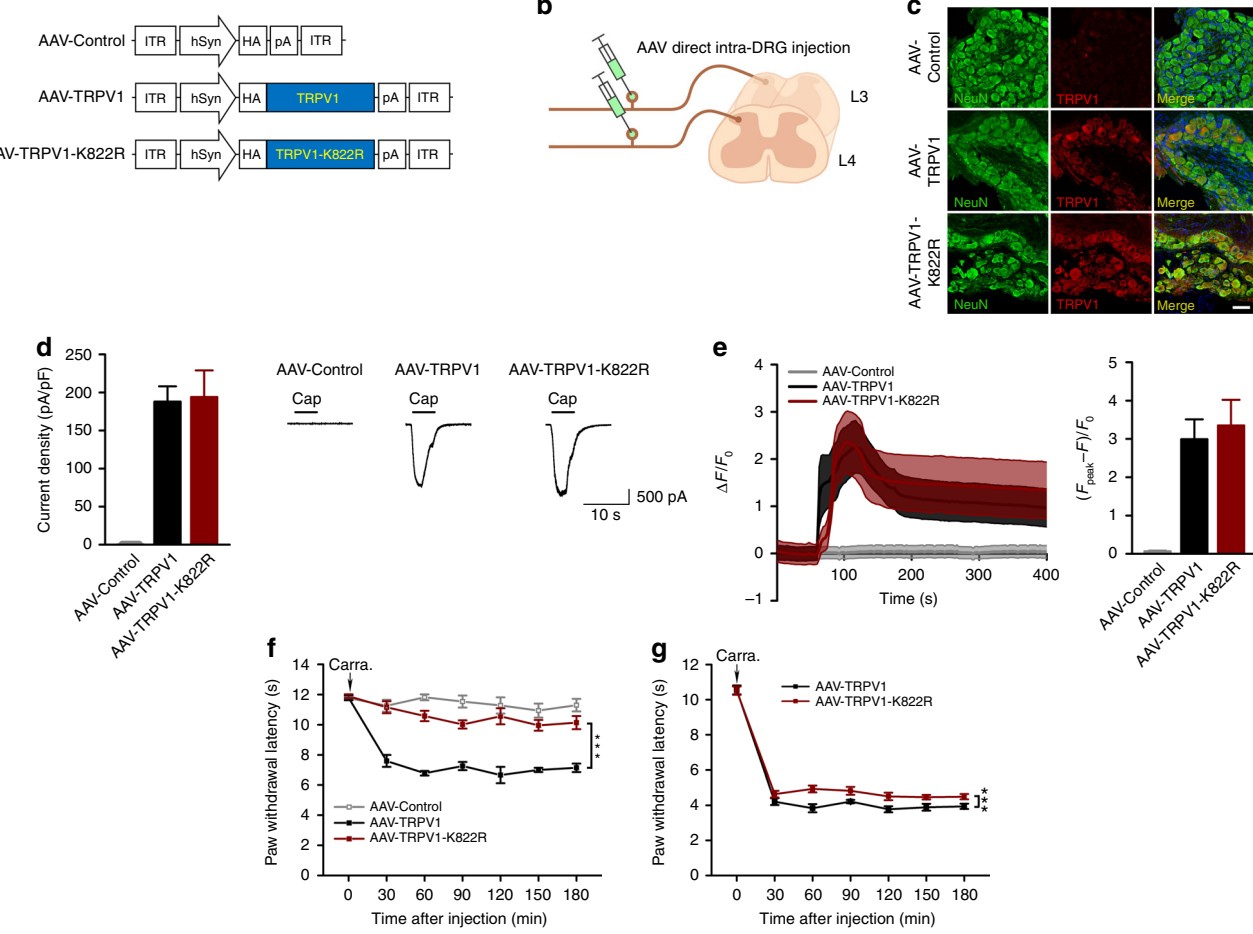

**Fig. 9** Essential role of TRPV1 SUMOylation in inflammatory thermal hyperalgesia. **a** Schematic diagram for the design of AAV-Control, AAV-TRPV1, and AAV-TRPV1-K822R. **b** Diagram for AAV viral infection into L3–L4 DRG of $TRPV1^{-/-}$ mice. The viruses (2 µl) were directly injected into the DRG. **c** The expression of TRPV1 in the left L3-L4 DRG of $TRPV1^{-/-}$ mice 4 weeks after the injection of AAV-Control, AAV-TRPV1, and AAV-TRPV1-K822R. DRG sections were stained with antibodies against TRPV1 (red) and NeuN (green). **d** Whole-cell currents at −60 mV evoked by capsaicin (1 µM) in neurons isolated from left L3-L4 DRG of $TRPV1^{-/-}$ mice that received the injection of AAV-Control, AAV-TRPV1, or AAV-TRPV1-K822R, as indicated, 4 weeks prior to DRG dissection. Representative traces (right) and quantification of current densities (left) are shown. The current densities are: AAV-Control, 0.58 ± 0.09 pA/pF ($n = 10$); AAV-TRPV1, 187.5 ± 18.5 pA/pF ($n = 11$); AAV-TRPV1-K822R, 194.3 ± 33.4 pA/pF ($n = 13$). **e** Capsaicin-evoked $Ca^{2+}$ transients in neurons as prepared in **d**. Cells were loaded with Fluo-4 and then images taken while capsaicin (1 µM) was applied at 60 sec. Time courses (left) and peak responses to capsaicin (right) are shown as $\Delta F/F_0$, where $F$ is fluorescence at a given time point, $F_0$ is the baseline fluorescence, and $\Delta F = F−F_0$. The peak values are: AAV-Control (gray), 0.06 ± 0.02 ($n = 21$); AAV-TRPV1 (black), 2.99 ± 0.52 ($n = 14$); AAV-TRPV1-K822R (wine) = 3.35 ± 0.67 ($n = 9$). **f** AAV-TRPV1-K822R failed to rescue inflammatory thermal hyperalgesia in $TRPV1^{-/-}$ mice. At 4 weeks after the injection of AAV-Control, AAV-TRPV1, and AAV-TRPV1-K822R in the left L3-L4 DRG, the $TRPV1^{-/-}$ mice were subject to the carrageenan edema model. The paw withdrawal latency (PWL) was measured as in Fig. 3e. Only AAV-TRPV1, but not AAV-Control or AAV-TRPV1-K822R, rescued thermal hyperalgesia induced by carrageenan administration to the hindpaws ($n = 5$ per group). ***$P < 0.0001$, AAV-TRPV1 vs. AAV-TRPV1-K822R. **g** AAV-TRPV1-K822R exhibited weak dominant negative effect on inflammatory thermal hyperalgesia in wild-type mice. Similar to **f** but AAV-TRPV1 and AAV-TRPV1-K822R were injected to the left L3–L4 DRG of wild type mice. At 4 weeks after the injection, mice were subject to the carrageenan edema model and PWL was measured by Hargreaves test ($n = 8$ per group). ***$P < 0.0001$, AAV-TRPV1 vs. AAV-TRPV1-K822R. Data are means ± s.e.m. Two-way ANOVA for **f** and **g**

*SENP1* cKO mice. As shown in Fig. 7i–l, similar to results obtained in CHO-K1 and HEK293T cells, augmenting SUMOylation in the sensory neurons by eliminating SENP1 also resulted in >50% decrease in the $Q_{10}$ value (Fig. 7k) and a left shift in the temperature response curve (Fig. 7l) without affecting the responses to capsaicin, acid and voltage (Fig. 8). Taken together, these results show that SUMOylation of TRPV1 selectively alters the temperature response of TRPV1 without affecting other mechanisms of TRPV1 gating.

**Lys-822 of TRPV1 is essential for thermal hyperalgesia.** To evaluate the contribution of SUMO1 modification at TRPV1-K822 in inflammatory thermal hyperalgesia, we ectopically

expressed wild-type TRPV1 and its K822R mutant in the DRG neurons of $TRPV1^{-/-}$ mice using adeno-associated viruses (AAV). A control AAV that contained empty vector (AAV-Control), AAV-TRPV1, or AAV-TRPV1-K822R, with the expression in neurons driven by the human synapsin promoter (hSyn) was injected into left L3–L4 DRG of adult $TRPV1^{-/-}$ mice (Fig. 9a, b). As shown by immunohistochemical labeling with the anti-TRPV1 antibody and anti-NeuN antibody, 4 weeks after direct intra-DRG injection, the expression of TRPV1 and TRPV1-K822R (red) was readily detectable in the left L3–L4 DRG neurons, but not AAV-Control injected ones (Fig. 9c).

To validate the functionality of the ectopically expressed channels, we performed whole-cell recording (Fig. 9d) and calcium

imaging (Fig. 9e) on freshly isolated DRG neurons that expressed TRPV1 or TRPV1-K822R. As expected, capsaicin (1 μM) evoked sizable inward currents at −60 mV in neurons that expressed TRPV1 or TRPV1-K822R with comparable current densities, but not in neurons from AAV-Control injected DRG (Fig. 9d). The agonist also elicited increases in intracellular $Ca^{2+}$ concentrations ($[Ca^{2+}]_i$) in neurons from DRG that were injected with AAV-

TRPV1 or TRPV1-K822R with comparable amplitudes, but not those from AAV-Control injected DRG (Fig. 9e). Therefore, the wild-type TRPV1 and its K822R mutant exhibited similar functional responses to the TRPV1 agonist, capsaicin, when ectopically expressed in DRG neurons of adult $TRPV1^{-/-}$ mice.

However, when the virus-injected mice were subjected to carrageenan injection into the left hindpaws, only the AAV-

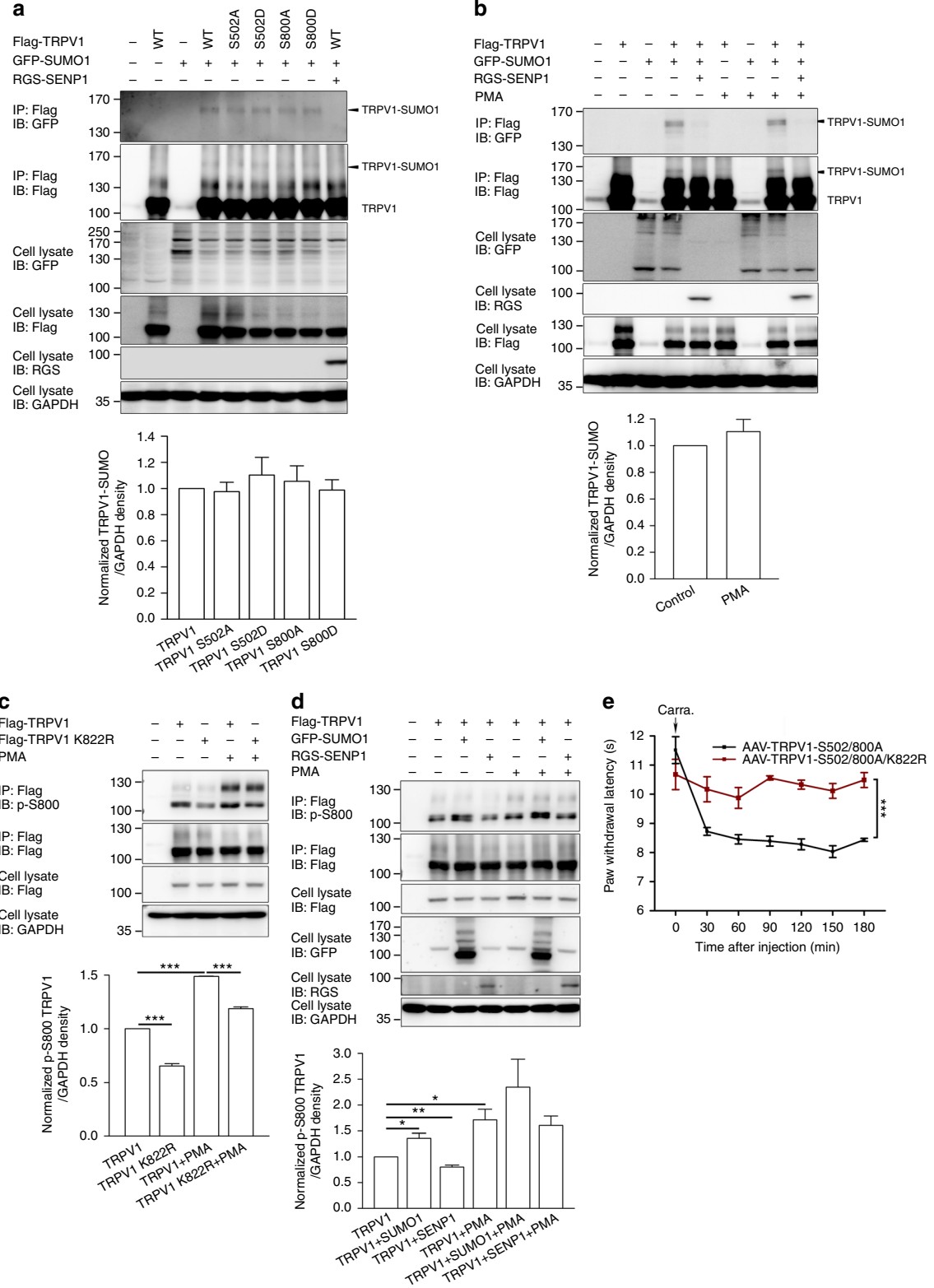

TRPV1 injected animals developed thermal hyperalgesia, as shown by the rapid decrease in paw withdrawal latency to heat stimulus within 30 min and the persistent decrease in the next 2.5 h (Fig. 9f). This mimicked the response of wild-type mice to the carrageenan injection (see Fig. 3e), demonstrating that the AAV-mediated ectopic expression of wild-type TRPV1 in DRG neurons of adult $TRPV1^{-/-}$ mice rescued inflammatory thermal hyperalgesia in the mutant animals. By contrast, the expression of TRPV1-K822R in DRG neurons of $TRPV1^{-/-}$ mice did not result in an enhanced thermal sensitivity in response to carrageenan injection, just like the AAV-Control infected (Fig. 9f) or the original uninfected $TRPV1^{-/-}$ mice (see Fig. 3e). Interestingly, the expression of TRPV1-K822R in DRG neurons of wild-type mice significantly reduced thermal hypersensitivity induced by carrageenan injection as compared to the infection by AAV-TRPV1, although the effect was moderate (Fig. 9g). This suggests a weak dominant negative effect of the mutant, and to abolish thermal hyperalgesia, all subunits of the TRPV1 channel, perhaps, need to be deSUMOylated. These findings strongly suggest that TRPV1 SUMOylation at K822 is required for the development of inflammatory thermal hyperalgesia.

Furthermore, since phosphorylation by protein kinase C (PKC) has been implicated in inflammatory factor-mediated potentiation of TRPV1 channel function[44,45], we asked whether there was a crosstalk between phosphorylation and SUMOylation in the development of inflammatory thermal hyperalgesia. PKC phosphorylates TRPV1 at two key Ser residues, S502 and S800[11,46,47], causing two separate effects on the channel: a decrease in $T_{threshold}$ and parallel increases in the sensitivities to other stimuli, including capsaicin, protons, and anandamide[48], and an increase in the trafficking of TRPV1 protein to the membrane surface[13]. To examine if PKC phosphorylation affects SUMOylation of TRPV1, we compared the levels of SUMOylated TRPV1 between CHO-K1 cells that expressed phosphorylation-deficient (S502A, S800A) and phosphorylation-mimetic (S502D, S800D) mutants of TRPV1. As shown in Fig. 10a, the SUMOylation level of TRPV1 was not altered by the mutations. Moreover, treatment with a PKC activator, phorbol 12-myristate 13-acetate (PMA, 1 µM, 15 min), did not affect TRPV1 SUMOylation (Fig. 10b). In contrast, SUMOylation enhanced PKC phosphorylation of TRPV1 (Fig. 10c, d). In CHO-K1 cells that expressed TRPV1-K822R, the level of phosphorylated TRPV1, as revealed using an antibody that recognizes phospho-S800 of TRPV1 (p-S800), was significantly lower than in cells that expressed wild-type TRPV1 under both the basal and PMA-stimulated conditions (Fig. 10c). The co-expression of GFP-SUMO1, but not RGS-SENP1, with wild-type TRPV1 also increased the phospho-TRPV1 levels under basal conditions (Fig. 10d). However, the PMA-stimulated TRPV1 phosphorylation was not significantly enhanced, despite the trend, by GFP-SUMO1 (Fig. 10d), probably because of the saturating effect of PMA on PKC.

The above data thus suggest that SUMOylation could enhance thermal sensitivity of TRPV1 through facilitation of PKC phosphorylation. To test if this were true, we used AAV to deliver the PKC phosphorylation deficient mutant of TRPV1, S502A/S800A, to DRG neurons of $TRPV1^{-/-}$ mice and then assessed inflammatory thermal hyperalgesia in the carrageenan edema model as described above. However, the TRPV1-S502A/S800A-injected mice exhibited similar paw withdrawal latency in response to radiant heat as the wild-type TRPV1-injected ones (Fig. 10e, comparing to AAV-TRPV1 in Fig. 9f), which was only eliminated by combining the K822R mutation with S502A/S800A (Fig. 10e). These results suggest that TRPV1 SUMOylation is essential for the development of inflammatory thermal hyperalgesia, and the response is not dependent upon TRPV1 phosphorylation, at least at S502 and S800.

## Discussion

The foregoing data demonstrate a mechanism by which the temperature sensitivity of TRPV1 channels is enhanced via SUMOylation of the TRPV1 protein at a C-terminal Lys residue. Temperature gating is an important feature of TRPV1, critical for the somatosensory response to noxious heat[1,3]. Therefore, modulation of temperature sensation represents a major mechanism underlying various forms of hyperalgesia. Intriguingly, the SUMOylation of TRPV1 is required for the development of inflammatory thermal hyperalgesia.

First, both SUMO1 and SENP1 are broadly expressed in DRG neurons. Second, TRPV1 is SUMOylated at K822 and this effect is enhanced in response to peripheral inflammation by carrageenan injection into mouse hindpaws. Third, SUMOylation specifically decreases the $Q_{10}$ and $T_{threshold}$ values for TRPV1 activation by heat, without affecting the channel's sensitivities to capsaicin, protons, and voltage, and the effect on temperature gating is eliminated by mutating lysine-822 into arginine. Fourth, conditional knockout of the deSUMOylation enzyme, SENP1, from sensory neurons not only exacerbates TRPV1 SUMOylation but also decreases $Q_{10}$ of heat-induced currents and aggravates inflammatory thermal hyperalgesia in both the carrageenan and CFA edema models. Last and most importantly, when introduced into DRG neurons via AAV-mediated infection, only wild-type TRPV1, but not its SUMOylation-deficient mutant (K822R), can rescue the impairment of inflammatory thermal hyperalgesia found in the $TRPV1^{-/-}$ mice and this effect was neither affected nor mimicked by preventing phosphorylation at S502 and S800 of TRPV1. Taken together, these results reveal a previously unknown function of SUMOylation in inflammatory pain,

**Fig. 10** Crosstalk between SUMOylation and PKC phosphorylation of TRPV1. **a** CHO-K1 cells were transfected with Flag-TRPV1 and its mutants together or not with GFP-SUMO1 or RGS-SENP1 as indicated. Cell lysates were subjected to IP by anti-Flag, followed by IB for GFP, Flag, and RGS. **b** CHO-K1 cells transfected with Flag-TRPV1, GFP-SUMO1, and RGS-SENP1 were untreated or treated with PMA (1 µM, 15 min) as indicated. Cell lysates were subjected to IP by anti-Flag, followed by IB for GFP, Flag, and RGS. **c** CHO-K1 cells transfected with Flag-TRPV1 and Flag-TRPV1-K822R were untreated or treated with PMA as indicated. To enhance the sensitivity, TRPV1 was enriched by IP using the anti-Flag antibody before IB was performed using the anti-phospho-S800 TRPV1 antibody (p-S800). ***$P < 0.0001$ for TRPV1 vs. TRPV1 K822R; TRPV1 vs. TRPV1+PMA; and TRPV1+PMA vs. TRPV1 K822R+PMA. **d** CHO-K1 cells transfected with Flag-TRPV1, GFP-SUMO1, and RGS-SENP1 were untreated or treated with PMA as indicated. The levels of phospho-S800 TRPV1 were determined as in **c**. Quantifications (means ± s.e.m.) of TRPV1 SUMOylation based on the results of IP Flag/IB GFP (**a**, **b**) and S800 phosphorylation based on the results of IP Flag/IB p-S800 (**c**, **d**) from three independent experiments are shown below the representative blots. **e** SUMOylation, but not PKC phosphorylation, at S502/S800 of TRPV1 is essential for inflammatory thermal hyperalgesia. Similar to Fig. 9f, but the left L3–L4 DRG of $TRPV1^{-/-}$ mice were injected with AAV-TRPV1-S502/800A ($n = 6$) or AAV-TRPV1-S502/800A/K822R ($n = 6$). The mice were subject to the carrageenan edema model at 4 weeks after the injection. The paw withdrawal latency was measured by Hargreaves test. AAV-TRPV1-S502/800A, but not AAV-TRPV1-S502/800A/K822R, rescued thermal hyperalgesia. ***$P < 0.0001$ for AAV-TRPV1-S502/800A vs. AAV-TRPV1-S502/800A/K822R. Data are means ± s.e.m. Student's $t$ test for **c** and **d**, two-way ANOVA for **e**

occurring through TRPV1 by lowering the $Q_{10}$ and $T_{threshold}$ of heat-evoked channel activation. The molecular mechanism demonstrated here adds a new dimension to TRPV1 regulation, which is already very complex. However, the specificity of SUMOylation on one single modality of this polymodal sensor, which is well known to respond to many forms of physical and chemical stimuli[49,50], is unique and its absolute requirement for mouse to develop thermal hyperalgesia in response to peripheral inflammation clearly distinguishes it from other types of PTM on TRPV1.

Previous studies have revealed that inflammatory mediators have multiple effects that may lead to nociceptor sensitization through augmentation of TRPV1 activity. Some are able to directly activate TRPV1, e.g., protons, or modulate the channel gating, e.g., prostaglandins and bradykinin[12,49,51], while others can increase the translocation of pre-existing TRPV1 to the plasma membrane (e.g., nerve growth factor (NGF), insulin and insulin-like growth factor 1)[12,13,49] or enhance the synthesis of TRPV1 protein (e.g., NGF)[14,52]. All these could contribute to hyperalgesia as they increase the overall TRPV1 channel activity. Among the PTM on TRPV1, phosphorylation is the best studied. Inflammatory mediators can sensitize TRPV1 via protein kinase A (PKA)[51,53], PKC[11,46,54,55], $Ca^{2+}$/calmodulin-dependent kinase II (CaMKII)[56], Src kinase[57], and cyclin-dependent kinase 5 (Cdk5)[38,58]. Among these, PKA and PKC phosphorylation has been widely investigated in various pain models[51,54]. Activation of PKC, particularly PKCε[59,60], strongly sensitizes the nociceptive response, which can be attenuated by both PKC inhibitors and mutations that disrupt the PKC phosphorylation sites on TRPV1[54]. In contrast, the $Ca^{2+}$-calmodulin-dependent phosphatase, calcineurin, is primarily responsible for channel desensitization[61], which is countered by PKA[51], a process facilitated by the PKA anchoring protein AKAP79/150[62]. Moreover, p38 mitogen-activated protein kinase is activated downstream from NGF to enhance TRPV1 protein expression and thereby help maintain the hypersensitivity of the sensory neurons. Thus, phosphorylation evoked by inflammatory mediators in general causes sensitization of the TRPV1 channel to multiple stimuli, including heat, which can account for inflammatory thermal hyperalgesia[49].

Our data reveal SUMOylation as a previously unknown form of PTM on TRPV1, which is augmented at least as quickly as within 1 h following the induction of peripheral inflammation in mice (Fig. 4c), specifically sensitizes the temperature response of the channel (Fig. 7), and critically contributes to the development of thermal hyperalgesia (Figs. 9f, 10e). Therefore, TRPV1 SUMOylation constitutes a key mechanism underlying peripheral sensitization to thermal stimuli under inflamed conditions. The results with S502A/S800A suggest that at least for these two PKC-regulated residues, phosphorylation is not required for thermal hypersensation. However, our data do not rule out the participation of other phosphorylated residues in the SUMO regulation of TRPV1 and the consequent development of inflammatory hyperalgesia. Interestingly, the naive SENP1 cKO animals did not exhibit spontaneous pain bouts despite a detectable increase in TRPV1 SUMOylation and enhanced thermal sensitivity in DRG neurons under basal conditions (Figs. 3d and 7j–l). There are at least two reasons for this. First, since SUMOylation is enhanced by inflammation, the changes brought about by the lack of SENP1 alone under the basal conditions probably were not enough to trigger pain in the cKO mice. Second, other consequences of inflammation, e.g., already decreased temperature threshold of TRPV1 activation due to a number of mechanisms associated with inflammation, such as acidic pH, phospholipase C activation[63], and changes in phosphorylation, etc., work in concert with SUMOylation to instigate thermal hyperalgesia.

SUMOylation has emerged as a critical PTM of nuclear and extranuclear proteins and a highly dynamic process, in which deSUMOylation is catalyzed by SENPs. This is also the case for TRPV1. In all examples where TRPV1 SUMOylation was detected, the overexpression of SENP1 but not the functionally dead SENP1 mutant, SENP1m, abolished the SUMOylation whereas the knockout of SENP1 in native DRG neurons led to enhanced TRPV1 SUMOylation. More importantly, conditional knockout of SENP1 in sensory neurons exacerbated inflammatory thermal hyperalgesia in both the carrageenan and CFA edema models (Fig. 3e, f), highlighting the role of SENP1 in suppressing the development of thermal hyperalgesia. Although SUMO1 conjugation was augmented for many different proteins in DRG upon peripheral inflammation, only TRPV1 appears to be critical for thermal hyperalgesia, as the SUMOylation-deficient TRPV1 mutant K822R failed to rescue inflammatory thermal hyperalgesia in $TRPV1^{-/-}$ mice while the SUMO-competent wild-type TRPV1 and S502A/S800A mutant did rescue (Figs. 9f,10e). Also notable is the observation that SENP1 cKO mice did not exhibit any other obvious neurological deficit, including normal nocifensive responses to capsaicin injection, suggesting that SENP1 specifically dampens thermal hyperalgesia under pathological conditions. This is relevant to pain management in a large array of human diseases where inflammation is a major contributor of hyperalgesia.

Based on the findings reported here, we propose a model for TRPV1 regulation of inflammatory thermal hyperalgesia through a pathway that involves SUMOylation of the channel protein (Supplementary Fig. 8). The SUMOylation status at K822 of TRPV1 is enhanced due to inflammation, which then lowers the threshold for channel activation by heat, resulting in enhanced sensitivity to thermal stimuli at the inflamed site. Remarkably, this previously unknown PTM of TRPV1 does not alter the expression of functional TRPV1 channels on cell surface, nor does it change the sensitivities of the channel to capsaicin, protons and voltage. However, despite the pronounced effect of the SUMOylation-deficient mutant on inflammatory thermal hyperalgesia, it cannot be ruled out that other changes associated with inflammation also participate in and serve to amplify the effect of SUMOylation. In such a case, SUMOylation may be considered a gate-keeper for the development of thermal hyperalgesia.

In summary, we demonstrate that TRPV1 SUMOylation is essential for the development of inflammatory thermal hyperalgesia, which occurs by acting at temperature gating of the channel. We show that peripheral inflammation results in hyperSUMOylation of TRPV1 in DRG neurons, which is antagonized with deSUMOylation by SENP1. We identified K822 at the C-terminus to be the critical residue for TRPV1 SUMOylation, which when mutated, can no longer mediate thermal hyperalgesia in response to inflammation. The new function demonstrated here for TRPV1 SUMOylation is important because it offers pivotal insights into the mechanism of inflammatory thermal hyperalgesia. The more in-depth understanding of the processes described here at the molecular level will represent an important step toward the development of more effective and specific pain therapies in future.

## Methods

**Antibodies, solutions and drugs**. Anti-CGRP was from Abcam (1;200, ab36001; Cambridge, MA, USA), anti-NF200 was from Sigma-Aldrich (1:200, N0142; Saint Louis, MO, USA), anti-peripherin was from Santa Cruz Biotechnology (1:300, SC7604; Dallas, Texas, USA), anti-TRPV1 was from Alomone (1:100 for Immunohistochemistry and 1:500 for western blot, ACC-030; Jerusalem, Israel) and Abcam (1:500 for western blot, ab203103; Cambridge, MA), anti-IB4 was from Fisher Scientific (1:300, I32450; Waltham, Massachusetts, USA), anti-p-S800-TRPV1 (1:1000, Abnova PAB8499; Taipei, Taiwan) and anti-NeuN was from Millipore (1:300, ABN78; Burlington, MA, USA). All other antibodies were as

described previously[24]. All compounds used in the $Ca^{2+}$ imaging and electro-physiological experiments were purchased from Sigma-Aldrich unless indicated otherwise. Collagenase type IA, trypsin from bovine pancreas type I, and deoxyribonuclease I from bovine pancreas (DNase I) were purchased from Sigma-Aldrich. Fluo-4 AM and Pluronic F-127 were purchased from Molecular Probes Inc (Eugene, OR, USA). Stock solutions of capsaicin (50 mmol $l^{-1}$), Fluo-4 AM (1 mmol $l^{-1}$), and Pluronic F-127 (20%) were prepared in anhydrous dimethyl sulf-oxide (DMSO). Collagenase (30 mg $ml^{-1}$), trypsin (12 mg $ml^{-1}$), and deoxyribonuclease I (3 mg $ml^{-1}$) were prepared in Hanks' balanced salt solution and stored at $-20\,^{\circ}C$. The stock was diluted to the desired concentration right before the experiment. Tamoxifen base in cornoil was prepared as described[29] and stored as 20 mg $ml^{-1}$ stock.

**Experimental animals.** All experimental protocols with animals were carried out in accordance with the guidelines for the Care and Use of Laboratory Animals of Shanghai Jiao Tong University School of Medicine and approved by the Institutional Animal Care and Use Committee (IACUC). Mice were housed under a 12-h/12-h dark/light cycle with ad libitum access to food and water. Mice were age- and sex-matched (males), and used at 8–12 weeks of age. The mouse strains used in this study were generated and maintained on the C57BL/6 background. $TRPV1^{-/-}$ mice have been described previously[64]. To create primary somatosensory neuron-specific SENP1 cKO mice ($SENP1^{flox/flox}$; $Prrxl1$-CreER$^{T2}$), $SENP1^{flox/flox}$[65,66] were crossed with $Prrxl1$-CreER$^{T2}$-Cre mice[29]. $SENP1^{flox/flox}$ mice produced in the same litter as the SENP1 cKO animals were therefore included as littermate controls unless indicated otherwise. When needed, tamoxifen was administered in a daily dose of 8 mg per 40 g of body weight for 4 consecutive days by oral gavage beginning from postnatal day 30 (P30). The mice were used for experiments 4 weeks after the drug administration.

**Mouse genotyping.** DNA was prepared from 0.5 cm of clipped tail specimen from mice and extracted in 20 μl of PBND lysis buffer containing 0.1 mg $ml^{-1}$ Proteinase K by incubation in a 55 °C thermomixer with vortexing for 2 h. Then the temperature was raised to 95 °C for 10 min to inactivate proteinase K. Genotyping was performed by PCR on the genomic DNA with 18 cycles of 95 °C for 2 min, 60 °C for 10 s, and 68 °C for 5 min. The primers for $SENP1^{flox/flox}$ were: loxP forward, 5′-AGAGTGAGACCCTGTCTCAACCCAAGC-3′ and loxP reverse, 5′-CACA-CAACTAAGTTAACTGCTGGAAACCAGAGC-3′, with the expected product sizes of 300 and 260 bps for $SENP1^{flox/flox}$ and wild-type mice, respectively. The primers for $Prrxl1$-CreER$^{T2}$ were: $Prrxl1$-CreER$^{T2}$ forward, 5′-TCGATGCAAC-GAGTGATGAG-3′ and $Prrxl1$-CreER$^{T2}$ reverse, 5′-TCCATGAGTGAAC-GAACCTG-3′, with the expected product size of 403 bps[29,66].

**cDNA constructs and mutagenesis.** GFP-SUMO1, His-SUMO1, and GFP:HA-ASIC3 were gifts from Guanghui Wang (Soochow University, Suzhou, China), Jianxiu Yu, and Ye Yu (Shanghai Jiao Tong University School of Medicine, Shanghai, China), respectively. RGS-SENP1, RGS-SENP1m, GFP-SENP1, and GFP-SENP1m plasmids were previously described[24]. To obtain Flag-TRPV1, the full-length TRPV1 (either rat or mouse) was cloned into p3 × FLAG-Myc-CMV-24 vector. Various Flag-TRPV1 mutations (K324R, K822R, S502A, S800A, S502D, S800D, S502/800A, and N604T) were generated using the QuikChange Site-Directed Mutagenesis Kit (Stratagene, La Jolla, CA).

**AAV virion injection into mouse DRG.** AAV virions were produced by Hanbio Biotechnology Co. Ltd. (Shanghai, China). Briefly, the HA tag (Control), HA-TRPV1, HA-TRPV1-K822R, HA-TRPV1-S502/800A, and HA-TRPV1-S502/800A/K822R were PCR-amplified individually and cloned using standard methods into pHBAAV-hSyn-mcs-WPRE vector. AAV-Control ($1.0 \times 10^{12}$ vg $ml^{-1}$), AAV-TRPV1 ($1.5 \times 10^{12}$ vg $ml^{-1}$), AAV-TRPV1-K822R ($1.7 \times 10^{12}$ per $ml^{-1}$), HA-TRPV1-S502/800A ($1.2 \times 10^{12}$ vg $ml^{-1}$), and HA-TRPV1-S502/800A/K822R ($1.2 \times 10^{12}$ vg $ml^{-1}$) were cotransfecting 10 μg pAAV-RC, 20 μg pHelper, and 10 μg of the desired pAAV plasmid into HEK293 cells with Lipofi-terTM (Hanbio Biotechnology Co.Ltd.). At 72 h after transfection, viral particles were collected by phosphate-buffered saline (PBS) and purified by the iodixanol step-gradient ultracentrifugation method. DRG injection of AAV was performed as described previously[67]. Briefly, mice were anaesthetized with pentobarbital. DRG (left L3–L4) were surgically exposed by dissecting and the viral solution (2 μl per DRG) was unilaterally injected into the DRG at a rate of 0.2 μl $min^{-1}$ with a glass micropipette connected to a Hamilton syringe controlled by a microsyringe pump controller (SYS-Micro4, WPI, Sarasota, FL, USA). The glass micropipette was removed after 10 min and the skin incision closed with wound clips. Mice were used for experiments at 4 weeks after the injection.

**Behavioral experiments.** Behavioral studies were performed with male mice of 8–12-week-old SENP1 cKO, littermate controls or $TRPV1^{-/-}$ mice. All behavioral training and tests were conducted during the light phase of the light/dark cycle by a trained observer blind to the genotype. Mice were habituated to the testing room for 60 min prior to all behavioral tests unless otherwise specified.

The open field was performed as previously described[66]. The test mouse was placed in the open field apparatus and allowed to freely explore for 30 min. Total distance traveled was recorded by Activity Monitor software (Med Associates, Inc. Fairfax, VT).

The rotarod test was performed as described previously[66]. Animals were trained on a rotarod with two practice sessions (0 rpm 30 s, 4 rpm 60 s, and 24 rpm 60 s) per day for 2 consecutive days. On the third day, the animal was habituated to rotation on the rod under a constant speed of 4 rpm for 30 s. Then the rotating speed was accelerated from 4 to 44 rpm for 300 s. The latency to fall from the rotating rod was recorded, with a cut-off time of 600 s. The protocol was repeated three times and the average latency of the three trials used for analysis.

The Hargreaves test was performed as described previously[68]. The test animal was placed in a clear plexiglass cylinder on top of a temperature-controlled Plantar Test Instrument (Ugo Basile 37370, Gemonio, Varese, Italy), which produces a high-intensity infrared light aimed at the plantar surface of the hindpaw. The withdraw latency of acute nocifensive response was determined by the onset of paw lift and/or lifting, licking and biting. Paw-withdrawal latency in response to heating was measured by a fixed infrared stimulus. Maximum stimulus duration was set at 20 s to prevent tissue damage. The animal was habituated in the plexiglass cylinder for 30 min. The baseline withdraw latency of the left hindpaw was measured three times with an interval of 15 min. For carrageenan-induced inflammation model, the left hindpaw was injected with 20 μl 2% (w/v) carrageenan. After 30 min, the withdraw latency was measured six times with a 30-min interval. Chronic/persistent inflammation was induced by CFA injection, as described previously[31]. Briefly, the left hindpaw of the test mouse was injected with 20 μl CFA (0.5 mg ml$^{-1}$, Sigma-Aldrich). The withdraw latency was measured at 2 h, 1 day, 3 days, 5 days and 7 days after CFA injection.

Capsaicin-induced nocifensive response was tested by injecting 10 μl capsaicin solution (0.2 μg μl$^{-1}$ in saline) into the left hindpaw of the test mouse. The animal was placed in a plexiglass cylinder. The total time spent on lifting, licking, and biting the injected paw was recorded during the first 10 min after injection[68,69].

**Cell culture and transfection.** Chinese hamster ovary cells (CHO-K1 line) were a kind gift from Professor Tong Zhou at the Department of Medicine (the University of Alabama School of Medicine, Birmingham, AL, USA) and HEK293T cell lines were purchased from the Chinese Academy of Science cell bank (Shanghai, China). CHO-K1 and HEK293T cells were cultured at 37 °C in a 5% $CO_2$ humidified incubator. CHO-K1 cells were grown in Ham's F12 medium containing 10% fetal bovine serum (FBS, Invitrogen, Carlsbad, CA). HEK293T cells were grown in DMEM (high glucose) supplemented with 10% FBS, 50 units $ml^{-1}$ penicillin, and 50 μg $ml^{-1}$ streptomycin. Cells grown into ~70% confluence were transfected with the desired DNA constructs using Lipofectamine 2000 (Invitrogen) following the protocol provided by the manufacturer.

**Preparation of DRG neurons.** Electrophysiological recordings and calcium imaging experiments were performed on dissociated DRG neurons. DRG were prepared as previously described with minor modifications[70,71]. Briefly, L3–L4 DRG were isolated from wild-type or SENP1 cKO mice, or $TRPV1^{-/-}$ mice following AAV injection, minced in cold Leibovitz's L-15 medium, and incubated for 60–90 min at 37 °C in Hank's balanced salt solution containing collagenase type IA (1 mg $ml^{-1}$), trypsin (0.4 mg $ml^{-1}$), and DNase I (0.1 mg $ml^{-1}$) and incubated at 37 °C for 30 min. After three washes in DMEM/F12 medium containing 10% FBS, 50 units $ml^{-1}$ penicillin, and 50 μg $ml^{-1}$ streptomycin, the cells were dispersed by gentle titration, plated on glass coverslips coated with 0.5 mg $ml^{-1}$ poly-D-lysine and 5 μg $ml^{-1}$ laminin, and cultured at 37 °C in 5% $CO_2$ overnight. Cells were used for $Ca^{2+}$ imaging and patch clamp recordings at 24 h post dissociation.

**Electrophysiology.** Patch-clamp recordings were made in the whole-cell configuration on both HEK293T and CHO-K1 cells transfected with the desired plasmid (s), and also on isolated mouse DRG neurons. For the recombinant expressing system, green fluorescence from GFP or GFP-SUMO1 was used as a marker for gene expression. Recording pipettes were pulled from borosilicate glass capillaries (World Precision Instruments, WPI) to give a resistance of 2–4 MΩ when filled with the pipette solution containing (in mM): 140 CsCl, 5 EGTA, and 10 HEPES, pH 7.4 adjusted with CsOH, and the bath solution for whole-cell recording from HEK293T/CHO-K1 cells contained (in mM): 140 NaCl, 5 KCl, 3 EGTA, and 10 HEPES, pH 7.4 adjusted with NaOH. For proton activation, the acidic solutions contained (in mM): 140 NaCl, 5 KCl, 5 EGTA, 1 $MgCl_2$, 10 glucose, and 10 MES, pH adjusted to desired values by NaOH. For recording under low pH conditions, the solution also contained 50 μM amiloride as a blocker to inhibit native acid sensing ion channels. For recording DRG neurons, the bath solution contained (in mM): 140 NaCl, 5 KCl, 2 $MgCl_2$, 2 $CaCl_2$, 10 glucose, 10 HEPES, pH 7.4 adjusted with NaOH and the pipette solution contained (in mM): 140 CsCl, 5 EGTA, and 10 HEPES, pH 7.2 adjusted with CsOH. Isolated cells were voltage clamped and held at $-60$ mV before recordings. Whole-cell currents were acquired at 5 kHz and filtered at 1 kHz using an EPC 10 amplifier and patchmaster software (HEKA, Lambrecht, Germany) or an Axopatch 200B amplifier (Molecular Devices, Sunnyvale, CA) and recorded through a BNC-2090/MIO acquisition system (National Instruments, Austin, TX) using homemade program, QStudio, which was designed by Dr. Feng Qin at State University of New York at Buffalo. Exchange of external solutions was performed using a gravity-driven local perfusion system. As determined by the conductance tests, the solution around the cell under study was fully

controlled by the application of a solution with a flow rate of 100 μl min$^{-1}$ or greater. For voltage dependence in the absence of an agonist, a voltage step protocol consisting of 200 ms depolarizing pulses from −100 to 200 mV with a 20-mV increment was triggered from the holding potential of −60 mV. The compensation of pipette series resistance and capacitance were taken by using the circuitry of the amplifier (>80%) to reduce voltage errors. Capsaicin stock was made in pure ethanol, capsazepine was dissolved in DMSO, and TEA-Cl was dissolved in distilled water. All the stocks were diluted in the bath solution to the desired concentrations right before the experiment. The final concentrations of ethanol or DMSO did not exceed 0.3%, which had no effect on the currents. All patch-clamp recordings were made at the room temperature (22–24 °C) except for temperature stimulation (see below). Electrophysiological data were analyzed offline with Clampfit (Molecular Devices), IGOR (Wavemetrics, Lake Oswego, OR, USA), SigmaPlot (SPSS Science, Chicago, IL, USA), and OriginPro (OriginLab Corporation, MA, USA). For concentration response analysis, the modified Hill equation was used: $Y = A1 + (A2−A1) \, / \, [1+10^{\wedge}(\log EC_{50}−X) \times n_{H}]$, in which $EC_{50}$ is the half maximal effective concentration, and $n_{H}$ is the Hill coefficient. $G–V$ curves for activation were derived from steady-state currents, converted to conductance and then fitted by the single Boltzmann function: $G/G_{max} = (1+\exp((V−V_{1/2})/\kappa))^{-1}$, where $V_{1/2}$ is the voltage at which the conductance ($G$) is half the maximum conductance ($G_{max}$), and $\kappa = RT/zF$ is the slope factor affecting the steepness of the activation, $R$ represents the gas constant, $T$ the absolute temperature, and $z$ the effective charge associate with voltage-dependent gating.

**Rapid temperature jump**. A single emitter infrared laser diode was used to produce temperature jumps, as previously described[71]. Briefly, a multimode fiber with diameter of 100 μm was used to transmit the launched laser beam. The other end of the fiber exposed the fiber core was placed close to the cells, the similar location as where the perfusion pipette is typically positioned. The laser diode was driven by a pulsed quasi-CW current power supply (Stone Laser, Beijing, China), and the pulsing of the controller was controlled from the computer through the data acquisition card by a custom made program, QStudio, which was kindly provided by Dr. Feng Qin at State University of New York at Buffalo. A blue laser line (460 nm) was coupled into the fiber to aid alignment. The beam spot on the coverslip was identified by illumination of GFP-expressing cells using the blue laser.

Constant temperature steps were generated by irradiating the tip of an open pipette filled with the bath solution and using the current of the electrode as the readout for feedback control. The laser was first powered on for 2 ms to reach the target temperature and then adjusted to maintain a constant pipette current. The profile of the modulation pulses was stored and subsequently played back to apply the temperature jump to the cell of interest. Temperature was calibrated offline from the electrode current based on the temperature dependence of electrolyte conductivity. The activation threshold of TRPV1 was determined as the temperature at which the slow inward current (~10% of its maximum response) was produced.

**Calcium imaging**. Calcium imaging was performed as previously described with minor modifications[72]. DRG neurons were loaded with the Ca$^{2+}$ indicator Fluo-4 AM (1 μM, diluted in extracellular solution with 0.01% Pluronic F-127) at 37 °C for 30 min. After dye loading, the cells were washed twice and coverslip was placed into the recording chamber and continuously perfused with extracellular solution at room temperature (21–25 °C). Cells were observed under an inverted microscope (DMI4000B, Leica, Wetzlar, Germany) while being excited with the 488 nm laser and then images taken by a CCD camera (DFC350 FX, Leica). Fluorescence intensity in individual neurons (region of interest) was recorded and analyzed by Leica Advanced Fluorescence Application software (AF6000; Leica). Drugs were applied by gravity via a microperfusion apparatus. Changes in cytosolic Ca$^{2+}$ levels were calculated by subtracting the fluorescence intensity at basal (mean value collected for 60 s prior to agonist addition) from that achieved after the agonist exposure.

**Immunohistochemistry**. Immunofluorescence staining was performed as described previously[73]. Mice were anesthetized with pentobarbital and perfused with 4% paraformaldehyde in PBS at 4 °C for 30 min. DRG were dissected, post-fixed in 4% paraformaldehyde at 4 °C for 2 h, and cryoprotected in 20% sucrose in PBS at 4 °C overnight. Before staining, 10 μm sections were post-fixed with 4% paraformaldehyde in PBS for 10 min and washed with PBST (PBS containing 0.1% TritonX-100). After blocking in 10% goat serum and 0.3% TritonX-100 in PBS for 1 h at room temperature, the sections were incubated at 4 °C overnight with primary antibodies. On the second day, the sections were washed in PBST three times and then incubated with desired secondary antibodies at room temperature for 1 h. The sections were then washed with PBST three times and mounted with Fluoromount-G (Southern biotech, Birmingham, Alabama, USA). Images were obtained using a Leica TCS SP8 confocal microscope system (Wetzlar, Germany).

**Immunoprecipitation and western blot**. Denaturing immunoprecipitations were carried out as previously described[74]. Briefly, denature lysis buffer I (50 mM Tris-HCl, pH 6.8, 2% SDS, 40 mM DTT, and 5% glycerol) was used to lyse the CHO-K1

or HEK293T cells at 24 h post-transfection and after washing with ice-cold PBS. The lysates were then incubated for 15 min at 95 °C before being diluted 10-fold using denature lysis buffer II (50 mM Tris-HCl, pH 7.4, 150 mM NaCl and 1% Nonidet P-40). After centrifugation for 8 min at 13,000×g and 4 °C, the supernatant of the lysates was transferred to a new tube, to which the desired primary antibodies were added. After an overnight incubation at 4 °C, A/G-Sepharose beads were added to the tube to purify proteins that had become associated with the antibodies. The beads were then centrifuged, washed three times, and boiled in the SDS sample buffer before being subject to SDS-PAGE and immunoblotting. Full images of western blots are shown in Supplementary Fig. 9.

**Surface biotinylation assay**. Surface biotinylation was performed following established protocols[24]. Cells were washed three times with ice-cold PBS (pH 8.0) supplemented with 1 mM MgCl$_2$ and 2.5 mM CaCl$_2$. Then Sulfo-NHS-LC-Biotin (Thermo Scientific, Waltham, MA, USA) was added to the same solution at 0.25 mg ml$^{-1}$ and incubated with cells at 4 °C for 30 min with gentle rocking. Unbound biotin group was quenched by the addition of 0.1 M glycine. Total proteins were extracted and incubated overnight at 4 °C with NeutrAvidin agarose beads (Thermo Scientific). The beads were washed three times with PBS (pH 8.0) and bound proteins were eluted with the boiling SDS sample buffer and used for immunoblotting.

**RNA extraction and analysis by RT-PCR**. RNA was extracted from the intact DRG using a Trizol RNA extraction kit (Tiangen, Beijing, China). RT-PCR was performed on 20 ng of total RNA with specific primers and a PrimeScript RT reagent Kit (Takara, Dalian, China) according to the manufacturer's protocol. The primers used for SENP1 cKO and its control mice were as follows: SENP1 forward (5′-GTGAAACGCTGGACAAAGAAG-3′), SENP1 reverse (5′-GTCTACAA-CAGCTAGACACCAG-3′); TRPV1 forward (5′-GGCTGTCTTCAT-CATCCTGCTGCT-3′), TRPV1 reverse (5′-GTTCTTGCTCTCCTGTGCGATCTTGT-3′); GAPDH forward (5′-CATGGCCTTCCGTATTCCTA-3′), GAPDH reverse (5′-GCCTGCTTCAC-CACCTTCTT-3′).

**Statistical analysis**. Fig. 7e, Supplementary Figs. 4t, 5t, 6t, and 7e are presented as mean ± s.d. All the other summary data are presented as mean ± s.e.m. with statistical significance assessed by Student's $t$ test for two-group comparison or two-way analysis of variance (ANOVA) tests for multiple group comparisons. Time course data from behavioral experiments were analyzed with two-way analysis of variance followed by Bonferroni's post-hoc to compare treatment effects over time. The value of $P < 0.05$ is considered to be statistically significant.

**Data availability**. The data that support the findings of this study are available from the corresponding author upon reasonable request.

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

## Acknowledgements

This study was supported by grants from the National Natural Science Foundation of China 31671053 (Y.L.), the National Basic Research Program of China 2014CB910300, the National Natural Science Foundation of China 31761163002 (Y.L.), 31371064 (Y.L.), 81171230 (Y.L.), 31230028 (T.-L.X.), 31671209 (J.Y.), and 31628005 (J.Y.), the China Postdoctoral Science Foundation (2016M601608 to Y.W.), the Natural Science Foundation of Hubei Province (2015CFA095 and 2017CFA063), Wuhan University funds (2042017kf0242 and 2042017kf0199 to J.Y.)  and the US National Institutes of Health (NS092377).

## Author contributions

Y.L. conceived the studies and designed the experiments. Y.W., Y.G., and Y.W. carried out the biochemical experiments and behavioral tests. Q.D. performed parts of the behavioral tests. T.Z., R.M., and H.L. performed parts of the biochemical experiments. Q.T., Q.L., K.M., and Y.W. performed the electrophysiological experiments and analyzed the results. H.L. and J.Y. assisted with data analysis. Y.D., W.R., J.C., T.-L.X., and M.X.Z. discussed the results. Y.D., W.R., J.C., and T.-L.X. provided technical support. Y.L., Y.W., J.Y., and M.X.Z. wrote the manuscript with inputs from all other authors. All authors discussed and commented on the manuscript.

## Additional information

**Competing interests:** The authors declare no competing interests.

