## [Peer Review File · Nature Communications]

Reviewers' comments:

Reviewer #1 (Remarks to the Author):

In this manuscript, Li and colleagues demonstrated for the first time that TRPV1 is sumoylated under inflammatory conditions. Moreover, they show that only the temperature-sensing modality of TRPV1 is affected by sumoylation. Finally, the authors define TRPV1 sumoylation site and provide both molecular and behavioral analysis of this process. The study is well designed and the manuscript is well written. Nevertheless, I have both major and minor issues the authors should address before this manuscript is ready for publication.

Major issues:

1) The authors used only the carrageenan inflammatory pain model throughout the manuscript. Although this model is excellent for describing acute inflammatory pain, it is not the 'gold standard' model for inflammatory pain, which include acute and chronic phases. The CFA model is regarded as the main model for inflammatory pain that simulates both the intensity and extent of the pain system response under inflammatory conditions. In addition, treatment with formalin evokes an acute inflammatory pain that directly involves nociceptors. Thus, I believe that including an analysis of TRPV1 sumoylation levels under these models (including in a time-dependent resolution) will enable the authors to claim its involvement in inflammatory pain (as stated in the title). In absence of these analyses, the authors should tune down the title and the remainder of the manuscript and only refer to the acute phase of inflammatory pain.

2) The authors do not show what exactly initiates TRPV1 sumoylation under inflammatory conditions. Moreover, the model in Figure 8 claims that TRPV1 has a basal sumoylation level. If this is the case, how do the authors explain that temperature threshold recorded on both cellular and behavioral levels remained constant without evidence of spontaneous pain bouts in naive animals? The authors should at least suggest a mechanism for the initiation of the sumoylation process of TRPV1 based on previous studies or provide an experimental evidence for how inflammation induces sumoylation in nociceptors.

3) This study is heavily relying on western-blot analyses. Although the authors are experts in this methodology, some features should be re-examined. For example, quantifications of western blot analysis are missing throughout the manuscript. This is mainly an issue in Figure 1 and 2 where neuronal cells are used. Additionally, different exposure times are used for gels presented in the same figure (e.g., Figure 2c and d). Finally, the authors rightfully claim that due to glycosylation of TRPV1 they cannot define if the band in 130kDa is sumoylated TRPV1 or not (Figure 2a). However, they are referring to this band and defining it as such in other parts of the paper (e.g., Figure 2c, d and supplement 4).

4) The authors 'jump' between recombinant systems and TRPV1 orthologues with no apparent reason. Why was the biochemistry done on CHO-K1 and the electrophysiology on HEK293 cells? Although both are heterologous systems, they are different in origin and characteristics. Why using mice for both ex vivo and in vivo analysis while using rat TRPV1 for the in vitro analysis? Although both are rodents, they are not identical. The authors should provide the reasoning beyond these choices.

5) The PKC experiments (supplement 4) should be presented in the result section and not in the discussion. PKC role in hyperalgesia is evident by more than 15 years of continued studies from multiple labs. The authors should use their elegant viral system (Figure 7) and inject a TRPV1(S502A/S800A) with and without the K822R mutation and describe the magnitude of the inflammatory pain evoked without PKC dependent TRPV1 phosphorylation.

Minor issues:

1) How does the co-expression of TRPV1 or TRPV1(K822R) with SUMO1 result in different Hill coefficients in the capsaicin dose-response compared to the dose responses without SUMO1 (lines

273-278)?

2) The recombinant expression of TRPV1 and the mutated receptor resulted in extraordinary large inward currents. These currents probably represent overexpression, which resulted in uncontrolled activation of the channel (hypersensitivity for agonists and disturbed kinetics). Please demonstrate that in this expression level capsaicin and protons evoked currents are outward-rectifying currents (as overexpression results in linear currents in voltage ramps). This point is important taking into account that the authors claim the sumoylation does not change TRPV1 sensitivity to capsaicin and protons, which may result from overexpression of the channel.

3) The temperature threshold should be extrapolated using the Arrhenius plot. In addition, please provide the Q10 of the different constructs.

4) In the introduction section, the reasoning for testing sumoylation of TRPV1 is not clear. However, the authors provide strong reasoning for this study in the result section. Please change the introduction to make it more easily to follow the reasoning beyond this study.

5) The authors use quite low dose of capsaicin in the behavioral analysis shown in Figure 3. Capsaicin was shown to evoke neurogenic inflammation and thermal hyperalgesia in higher doses. The authors should state the reasoning for using capsaicin at such low doses.

Reviewer #2 (Remarks to the Author):

This is an interesting paper showing that sumoylation of TRPV1 has an effect on temperature gating of the channel and contributes to heat hyperalgesia. The effects shown seem robust and important. The identification of the sumoylated residue on TRPV1 is a very novel modification of the protein and the rescue experiment done using viral vectors and the mutated sumoylated lysine is a powerful experiment. I have a few suggestions to improve the paper.

1) The effects are only shown in one inflammation model. Does sumoylation of TRPV1 change in other models (e.g CFA or NGF)? How generally applied is this model?

2) The discussion is really quite long. I found it to be overly verbose, in particular the idea of gating for phosphorylation, which is highly speculative.

3) the data in DRG with carrageenan in the paw in figure 1 is interesting but the TRPV1 population that must lead to thermal hyperalgesia at this early time point is in the nerve terminals in the paw and not in the DRG cell bodies. Can the authors provide evidence that there is an increase in SUMO modifications at the site of inflammation at this early time point?

4) I am a little confused on the reasoning in the SENP1 cKO experiment. Since the cKO of SENP1 increases global SUMO1 conjugation in the DRG, why does this not induce thermal hyperalgesia by itself, especially considering that this is a conditional mutation. Does TRPV1 SUMO1 conjugation only change with inflammation while many other proteins change simply by cKO of SENP1? Figure 4 seems to argue against this idea. Is there a location-specific change with more distal changes in nerve terminals not occurring with cKO of SENP1?

5) in the discussion, allodynia may not be best explained by sensitization of nociceptors so the authors should be cautious with their language there.

Reviewer #3 (Remarks to the Author):

The paper by Li and co-workers identifies SUMOylation as a novel post-translational modification of the TRPV1 channel and describes a role for SUMOylation of TRPV1 at lysine residue 822 in the development of thermal hyperalgesia in an inflammatory state. An outstanding finding is the demonstration that TRPV1 knockout mice, which normally do not develop inflammatory thermal hyperalgesia, become hyperalgesia upon "add-back" of a virus overexpressing TRPV1 but fail to

develop hyperalgesia when a SUMOylation-deficient TRPV1 channel is added into these mice. The novel findings are based on in vitro and in vivo studies using cell lines, sensory neurons and conditional knockout mice, which collectively support the main findings of the paper. Whilst the behavioral and electrophysiological studies are mostly of good quality and high rigor, the biochemistry is lacking important controls and additional conditions to support the conclusions. Further, the potential cross-talk between other modifications (glycosylation, phosphorylation) of TRPV1 and SUMOylation of TRPV1 is only superficially addressed and needs further development. The statistics and the references cited are appropriate, although specific examples of SUMOylation would be preferred over general reviews.

Specific comments and suggestions for improvement:

1. Abstract: the statement that SUMO1 and SENP1 are the major regulators of protein SUMOylation is an over-interpretation and should be restated.
2. As written, the Introduction is vague. For example, instead of using several reviews to cite the importance of SUMOylation in the nervous system – none of which pertain to the pain pathway – perhaps the authors could instead give examples of how, in the nociceptive pathway, protein targets of SUMOylation are involved.
3. Why is SUMOylation increased in carrageenan-treated tissues (Fig. 1d, e)? What are some potential reasons that this might happen? And how might this be relevant? Why are there no SUMO1-conjugated proteins in the region between 40 to 90 kD in the blots shown in panels d and e of Fig 1? Typically in a cell, there is a high number of proteins of sizes between 40 to 90 kD.
4. The data in Fig. 4f supporting lys 822 contributing to TRPV1 SUMOylation are convincing. However, the data using native conditions to prove SUMOylation of TRPV1 need additional controls in order to conclude that TRPV1 is being SUMOylated. In Fig 4a, how do the authors conclude that the band at ~170 kD in the SUMO1 blot (top blot) is actually TRPV1 and not another protein of similar size? From the cell lysates shown below, it is clear that there are several proteins in this vicinity. In Fig. 4c, d and f, there are no negative controls for the IP experiments. The authors ought to repeat the experiments immunoprecipitating TRPV1 in denaturing conditions and probing with SUMO1 antibody to convincingly demonstrate that TRPV1 is SUMOylated. Importantly, in all of figure 4 panels, quantification is missing and should be provided. Individual samples values would be useful to report as levels of SUMO seems to be different between samples within the same experiment (Figure 4b).
5. The recordings in Figures 5 and 6 are of good quality but are missing expression of Ubc9. This is critical, as in our experience HEK 293 cells do not express adequate machinery for SUMOylation. Therefore, the experiments in Fig. 5 and 6 (particularly TRPV+SUMO1 and TRPV1-K822R+SUMO1 conditions) must be redone in the presence of over-expressed Ubc9. Note that the authors used an overexpression strategy in Fig. 4f to boost SUMOylation. The authors must also explain and justify their use of CHO-K1 cells for biochemistry and HEK293T cells for electrophysiology. Results need to be reproduced in DRG neurons.
6. Fig 7 contains of the two most interesting features of this work: behavioral demonstration of the importance of SUMOylation. It would be important to evaluate behavior in wildtype mice following the introduction of the TRPV1 K822R SUMO-mutant as this may reveal whether the loss of SUMOylation is a dominant negative phenotype. The temperature experiments are very well done and highlight the other salient feature of this paper: the ~7 C shift in thermo sensation. Again, as above, at least for the TRPV+SUMO1 and TRPV1-K822R+SUMO1 conditions, the experiments ought to be redone in the presence of over-expressed Ubc9. Results need to be reproduced in DRG neurons.
7. Is TRPV1 phosphorylation changing the SUMOylation status? Which modification is more or most

important? The cross-talk between modifications is not sufficiently investigated. The data in Supplementary Fig. 4a, for example, shows that there is some change in PKC phosphorylation in the TRPV1 mutant but no data is provided quantifying it. The cross-talk between modifications is an interesting aspect of regulation of proteins and may be the case here for TRPV1 channels, too.

Minor suggestions:

1. In the manuscript sumoylation/desumoylation should be written SUMOylation/deSUMOylation since SUMO is an acronym.
2. Product numbers for all materials should be provided.
3. Details on virus production and titration are missing.
4. Need to control for loading in SUMO IP experiments
5. Please provide a neuronal marker staining for figure 7C.
6. No negative control for figure sup 1
7. Supplementary Figure 2, flag sumo is not in the right place. Authors are expecting a higher shift than the experiment is suggesting. Also, no negative control provided.
8. No negative control for surface expression provide in Supplementary figure 3.
9. A general suggestion in some of the biochemistry experiments would be to use the GFP antibody instead of the SUMO1 antibody.

Authors' Responses to the Referees' Comments:

Referee #1 (reviewers' comments in *italic*):

In this manuscript, Li and colleagues demonstrated for the first time that TRPV1 is sumoylated under inflammatory conditions. Moreover, they show that only the temperature-sensing modality of TRPV1 is affected by sumoylation. Finally, the authors define TRPV1 sumoylation site and provide both molecular and behavioral analysis of this process. The study is well designed and the manuscript is well written. Nevertheless, I have both major and minor issues the authors should address before this manuscript is ready for publication.

Response: We sincerely thank the knowledgeable reviewer for his/her valuable time, critical comments and insightful suggestions that have helped us clarify and strengthen our main findings. We have conducted new experiments and added experimental details to the results section to address issues raised by the reviewer.

Major issues:

1) The authors used only the carrageenan inflammatory pain model throughout the manuscript. Although this model is excellent for describing acute inflammatory pain, it is not the 'gold standard' model for inflammatory pain, which include acute and chronic phases. The CFA model is regard as the main model for inflammatory pain that simulates both the intensity and extent of the pain system response under inflammatory conditions. In addition, treatment with formalin evokes an acute inflammatory pain that directly involve nociceptors. Thus, I believe that including an analysis of TRPV1 sumoylation levels under these models (including in a time-dependent resolution) will enable the authors to claim its involvement in inflammatory pain (as stated in the title). In absence of these analyses, the authors should tune down the title and the remainder of the manuscript and only refer to the acute phase of inflammatory pain.

Response: We appreciate the thoughtful suggestions of the reviewer. As suggested, we have added the CFA model to test the involvement of TRPV1 SUMOylation in regulating nociceptive signaling of inflammatory pain.

As shown in new Fig. 3f, mice that received unilateral intraplantar administration of Complete Freund's Adjuvant (CFA) into the left hind paws developed thermal hyperalgesia at the injected paws by 2 hrs after injection, which persisted for 7 days throughout the testing period. Thermal hyperalgesia was indicated by a significant reduction in the paw withdrawal latency (PWL) in response to an infrared heat stimulation post-CFA when compared with the test on the uninjected paws. Bonferroni post test showed that sensory

thresholds at the ipsilateral side of the CFA-treated mice were significantly different from contralateral side hindpaw at all the time points. Importantly, the SENP1 cKO mice exhibited a stronger reduction of PWL than their control littermates, reaching the lowest value of 2.32 ± 0.14 s at 1 day after CFA injection, which remained lower than wild type control throughout the 7 days test period (Fig. 3f). For comparison, the wild type control mice displayed a lowest PWL of 3.32 ± 0.27 s (Fig. 3f). Therefore, the reduced ability to regulate protein SUMOylation and deSUMOylation in DRG neurons because of the SENP1 deficiency exacerbated inflammatory thermal hyperalgesia in both the carrageenan and CFA models.

We agree with the reviewer's point that treatment with formalin evokes an acute inflammatory pain that directly involve nociceptors. However, we are unable to perform these experiments due to time constraints and the complication that formalin directly activates a related channel, TRPA1.

2) *The authors do not show what exactly initiate TRPV1 sumoylation under inflammatory conditions. Moreover, the model in Figure 8 claims that TRPV1 has a basal sumoylation level. If this the case, how the authors explain that temperature threshold recorded on both cellular and behavioral levels remained constant without evidence of spontaneous pain bouts in naïve animals? The authors should at least suggest a mechanism for the initiation of the sumoylation process of TRPV1 based on previous studies or provide an experimentally evidence for how inflammation induces sumoylation in nociceptors.*

Response: We thank the reviewer for raising these important and interesting points. We tested the expression of SENP1 and Ubc9 in DRG and spinal cord after carrageenan injection, and found no difference between the contralateral and ipsilateral side (see Reviewer Fig. 1). SUMOylation is a highly dynamic and spatially defined process that regulates multiple cellular events and requires coordinated activities of the SUMOylation machinery: activating enzyme (E1), conjugating enzyme (E2), SUMO ligases (E3) and deSUMOylating isopeptidases that cooperate with SUMO-targeted ubiquitin ligases. Because of its dynamic nature, SUMOylation is difficult to detect. Typically, only a very small portion of the substrate protein is SUMOylated at any given moment. In addition, there is not a known ligand or cue (either endogenous or exogenous) that can dramatically boost SUMOylation levels, making it hard to deduce the precise cellular mechanism(s) that initiate SUMOylation. Despite these, it has been previously reported that inflammatory mediators increase SUMOylation of retinoid X receptor α , a critical heterodimeric partner of PXR (Schneider Aguirre and Karpen, 2013), suggesting that some constituents of inflammation can enhance SUMOylation. Thus, the balance between SUMOylation and deSUMOylation can be shifted towards SUMOylation under inflammation. Our data suggest that the lack of SENP1 in sensory neurons further exacerbates such a shift, leading to

enhanced thermal hyperalgesia. Notably, the enhanced SUMOylation in DRG from SENP1-cKO animals under basal conditions (Fig. 2c) is not as great as the carrageenan-induced increase (Fig. 1e), supporting that inflammation promotes SUMOylation in DRG neurons.

We believe that there are at least two reasons for the lack of spontaneous pain bouts in naïve SENP1-cKO animals. First, SUMOylation did not reach high enough level under the basal conditions to trigger pain in the cKO mice. Second, other consequences of inflammation, e.g. already decreased temperature threshold of TRPV1 activation due to a number of mechanisms associated with inflammation, such as acidic pH, PIP₂ breakdown and changes in phosphorylation, etc., work in concert with SUMOylation to support thermal hyperalgesia. The latter would argue that enhanced SUMOylation alone in DRG neurons, with the few °C drop in temperature threshold of TRPV1 activation (Fig. 6i-l), may be insufficient to cause pain. We have revised Fig. 6 and Supplemental Fig. 8 and the text of Discussion to include these points (Page 17, line 24 to Page 18, line 19).

References:

Schneider Aguirre R & Karpen SJ. Inflammatory mediators increase SUMOylation of retinoid X receptor α in a c-Jun N-terminal kinase-dependent manner in human hepatocellular carcinoma cells. *Mol Pharmacol.* 2013 84(2):218-26.

3) This study is heavily relying on western-blot analyses. Although the authors are experts in this methodology, some features should be re-examined. For example, quantifications of western blot analysis are missing throughout the manuscript. This is mainly an issue in Figure 1 and 2 where neuronal cells are used. Additionally, different exposure times are used for gels presented in the same figure (e.g., Figure 2c and d). Finally, the authors rightfully claim that due to glycosylation of TRPV1 they cannot define if the band in 130kDa is sumoylated TRPV1 or not (Figure 2a). However, they are referring to this band and defining it as such in other parts of the paper (e.g., Figure 2c, d and supplement 4).

Response: We thank the referee for raising a number of important points. To address the reviewer's questions, we have now performed several additional experiments.

First, as requested by the reviewer we have provided quantification of the biochemical analyses to bolster our conclusions throughout and included statistical analysis in the revised Fig. 4, 5, 8 and Supplemental Fig. 1.

Second, we have performed additional western blotting experiments using nearly identical exposure times and included quantifications of the new western blot data in the revised manuscript. Due to the dynamic nature of SUMOylation and the very low level (often less than 1% under basal conditions)

of a particular protein pool being SUMOylated at any given time, we have to use very long exposure times in order to reveal the bands for some experiments.

Third, we have examined the relationship between SUMOylation and glycosylation of TRPV1 by experiments using glycosidase PNGase F and a glycosylation-deficient TRPV1 mutant, N604T (Jahnel et al., 2001; Wirkner et al., 2005). Our new results suggest that the SUMO-labeled TRPV1 band, which we have been looking at, is not affected by glycosylation (Supplemental Fig. 1). Therefore, most likely, only the non-glycosylated TRPV1 is examined here for SUMOylation. Note that glycosylated TRPV1 proteins appear as diffused higher molecular weight proteins of lower abundance than the non- or incompletely glycosylated ones. Since it is already very difficult to detect the SUMOylated non-glycosylated TRPV1 because of low abundance, the SUMOylation of higher molecular weight glycosylated TRPV1 species would be even harder to detect given the heterogeneity in size and poor resolution of the gel in the high molecular weight range.

Finally, we apologize for not addressing clearly the difference in SUMO1 between experimentations using heterologous and native systems in the original manuscript. SUMO tag has an expected molecular weight of 12 kD and migrates at ~15-18 kD. TRPV1 is predicted to be approximately 100.7 kD. Therefore, the SUMOylated TRPV1 would be approximately 120 kD. However, TRPV1 is glycosylated in heterologous expression systems, yielding bands in the range of ~125-135 kD, which would interfere with the detection of SUMOylated TRPV1. Therefore, we used GFP-SUMO1, instead of SUMO1, in heterologous systems. The GFP adds additional weight to give an expected size of ~160 kD, away from the glycosylated TRPV1. In the native DRG, we could only work with endogenous SUMO1. However, since endogenous TRPV1 in freshly isolated DRG (containing mainly neuronal somata) was found to be barely glycosylated (Kedei et al., 2001; Xing et al., 2012), this is not a problem. In addition, we used PNGase F to deglycosylate TRPV1, then used IP to detect the SUMOylation of TRPV1 to make sure the band is not glycosylated TRPV1 (Reviewer Fig. 4). Therefore, we could safely refer the ~125 kD band as SUMOylated TRPV1 in Fig. 4c.

References:

Kedei N, Szabo T, Lile JD, Treanor JJ, Olah Z, Iadarola MJ & Blumberg PM (2001) Analysis of the native quaternary structure of vanilloid receptor1. *J. Biol. Chem.* **276**, 28613-28619.

Xing BM, Yang YR, Du JX, Chen HJ, Qi C, Huang ZH, Zhang Y & Wang Y (2012) Cyclin-dependent kinase 5 controls TRPV1 membrane trafficking and the heat sensitivity of nociceptors through KIF13B. *J Neurosci.* **32**, 14709-14721.

Jahnel R, Dreger M, Gillen C, Bender O, Kurreck J & Hucho F (2001)

Biochemical characterization of the vanilloid receptor 1 expressed in a dorsal root ganglia derived cell line. *Eur. J. Biochem.* **268**, 5489-5496.

Wirkner K, Hognestad H, Jahnel R, Hucho F & Illes P (2005) Characterization of rat transient receptor potential vanilloid 1 receptors lacking the N-glycosylation site N604. *Neuroreport* **16**, 997-1001.

4) The authors 'jump' between recombinant systems and TRPV1 orthologues with no apparent reason. Why the biochemistry was done on CHO-K1 and the electrophysiology on HEK293 cells? Although both are heterologous systems, they are different in origin and characteristics. Why using mice for both ex vivo and in vivo analysis while using rat TRPV1 for the in vitro analysis? Although both are rodents, they are not identical. The authors should provide the reasoning beyond these choices.

Response: The "jump" between the two systems was a result of collaboration between the two main laboratories; both are deeply involved in the project but each has their preferred systems to work with. Thanks to the reviewer's comments, we have performed a new series of experiments and confirmed the key results in both HEK293 and CHO-K1 cells and with both rat and mouse TRPV1.

First, we have performed additional biochemistry experiments in HEK293 cells and electrophysiology experiments in CHO-K1 cells. The new results reproduced the previous biochemical and electrophysiological findings in a different recombinant system, respectively. These data are included in the revised Fig. 5, Fig. 6, Supplemental Fig. 2 and Supplemental Fig. 5.

Second, we have repeated in vitro experiments using mouse TRPV1 to examine if SUMOylation affects the channel's temperature response. The results are similar to that obtained using rat TRPV1. These data are included in the revised Supplemental Fig. 5 and Supplemental Fig. 7.

5) The PKC experiments (supplement 4) should be presented in the result section and not in the discussion. PKC role in hyperalgesia is evident by more than 15 years of continued studies from multiple labs. The authors should use their elegant viral system (Figure 7) and inject a TRPV1(S502A/S800A) with and without the K822R mutation and describe the magnitude of the inflammatory pain evoke without PKC dependent TRPV1 phosphorylation.

Response: We thank the referee for raising this important point. To address this question, we have performed the following experiments.

First, we have performed new experiments to examine TRPV1 phosphorylation in the absence and presence of SUMOylation. The new

results are incorporated into previous Supplemental Fig. 4 and the figure has been moved to the main part of the manuscript as revised Fig. 8.

Second, as suggested by the reviewer, we have used the viral delivering system to introduce TRPV1(S502A/S800A) with and without the K822R mutation to DRG neurons of the *Trpv1* null mice and then assessed inflammatory thermal hyperalgesia in the carrageenan edema model. The TRPV1(S502A/S800A)-injected mice exhibited similar PWL in response to radiant heat as the wild type TRPV1-injected ones (Fig. 8e), while the combination with K822R, TRPV1(S502A/S800A/K822R), eliminated such a response (Fig. 8e). These results suggest that TRPV1 SUMOylation is essential for the development of inflammatory thermal hyperalgesia, and the response is not dependent upon TRPV1 phosphorylation, at least at S502 and S800.

Minor issues:

1) How the co-expression of TRPV1 or TRPV1(K822R) with SUMO1 resulted in different Hill coefficients in the capsaicin dose-response compared to the dose responses without SUMO1 (lines 273-278)?

Response: Thanks. The different Hill coefficients in the capsaicin dose-response was caused by the auto-fitting weight. We had fixed the n_H values of TRPV1 with SUMO1 and refit the plot, and the result trace can cover all the dots. These suggest that SUMOylation of TRPV1 does not affect its sensitivity to capsaicin (Supplemental Fig. 9).

Following the suggestion of the reviewer and also the recommendation of reviewer 3, we have repeated the experiments in both HEK293 and CHO-K1 cells using both mouse and rat TRPV1 constructs and included Ubc9 to facilitate SUMOylation. We have used new data to create the figures, which do not show different Hill coefficients in the capsaicin dose responses.

2) The recombinant expression of TRPV1 and the mutated receptor resulted in extraordinary large inward currents. These currents probably represent overexpression, which resulted in un-controlled activation of the channel (hypersensitivity for agonists and disturbed kinetics). Please demonstrate that in this expression level capsaicin and protons evoked currents are outward-rectifying currents (as overexpression results in linear currents in voltage ramps). This point is important taking into account that the authors claim the sumoylation does not change TRPV1 sensitivity to capsaicin and protons, which may result from overexpression of the channel.

Response: We have performed voltage ramps from -100 mV to +100 mV on recombinant systems expressing TRPV1 in the absence and presence of 0.03 μ M capsaicin. As shown in Fig. 5a-d, TRPV1 exhibited strong outward rectification. The co-expression of SUMO1 did not affect the outward

rectification, which would indicate that under our experimental conditions, TRPV1 did not show hypersensitivity to agonists and disturbed kinetics.

The shape of the I-V for TRPV1 changes according to the strength of the stimulation, consistent with the idea that ligand stimulation alters the voltage dependence of the channel (Voets et al., 2004). Therefore, at high capsaicin concentrations or very low pH, the inward TRPV1 current can be quite big. DRG neurons also show large inward current in response to high capsaicin concentrations and the I-V at negative potentials can be linear. Therefore, outward rectification is not a characteristic feature of TRPV1 currents. Nonetheless, we have now provided example I-V curves obtained by voltage-ramps as requested by the reviewer as explained above (new Fig. 5a-d, Supplemental Fig. 5a-d, Supplemental Fig. 6a-d and Supplemental Fig. 7a-d).

Reference:

Voets T, Droogmans G, Wissenbach U, Janssens A, Flockerzi V, Nilius B. (2004) The principle of temperature-dependent gating in cold- and heat-sensitive TRP channels. *Nature* 430(7001):748-754.

3) The temperature threshold should be extrapolated using the Arrhenius plot. In addition, please provide the Q10 of the different constructs.

Response: These have been corrected as suggested. Both Arrhenius plots and Q10 values are now shown (new Fig. 5f and k, Supplemental Fig. 5u, Supplemental Fig. 6u and Supplemental Fig. 7u).

4) In the introduction section, the reasoning for testing sumoylation of TRPV1 is not clear. However, the authors provide strong reasoning for this study in the result section. Please change the introduction to make it more easily to follow the reasoning beyond this study.

Response: Following the reviewer's suggestion, we have extensively revised the introduction.

"TRPV1 is a ligand-gated non-selective cation channel prominently expressed in small- and medium-diameter sensory neurons within dorsal root ganglia (DRG), where it functions as a sensor for an array of exogenous and endogenous chemical and physical stimuli, including capsaicin, noxious heat (> 43 °C) and acidosis (pH < 5.9)¹⁻³. Activation of TRPV1 by noxious stimuli induces inward cationic currents and the resulting action potentials in nociceptive DRG neurons then convey nociceptive information to the spinal dorsal horn⁴⁻⁶. Thus, TRPV1 is considered to be one of the major contributors of nociception⁷⁻⁹.

Thermal hyperalgesia represents a pathological state in which the threshold of pain sensation to a thermal stimulus is decreased. Because of the central involvement in the thermal hyperalgesia that accompanies inflammatory pain,

TRPV1 has been the focus of intensive research aimed at understanding the mechanism underlying inflammatory thermal hyperalgesia¹⁰⁻¹². It is well known that TRPV1 function is potentiated by inflammatory mediators, which act through receptor-operated intracellular signaling cascades to modify channel gating^{11,13-15}. For instance, pro-inflammatory sensitization of TRPV1 involves modifications by protein kinases, leading to a decrease in activation threshold and hence an augmentation of channel activity¹⁶⁻¹⁹. However, the potentiation of TRPV1 can additionally be achieved by a rapid increase in its cell surface expression via translocation from a subcellular vesicular reservoir to the plasma membrane²⁰⁻²², as well as the transcriptional and translational control of TRPV1 expression²³⁻²⁶.

While phosphorylation represents the best studied inflammation-related post-translational modification (PTM) mechanisms that exert functional modulation via either changes in plasma membrane expression or alterations in the biophysical properties of the TRPV1 channel, the effects of other types of PTM on this channel and their implications in inflammatory thermal hyperalgesia remain largely unexplored. As a form of PTM, SUMO modification has recently emerged as a key regulatory pathway of many biological processes²⁷⁻²⁹. SUMOylation modifies protein function by covalently binding a member of the SUMO family to the target protein. Such a modification can facilitate or prevent inter- and intra-molecular interactions via conformational changes or direct steric hindrance. An increasing number of ion channels, including GluK2-containing kainate receptors³⁰, voltage-gated potassium channels, Kv2.1³¹ and Kv1.5³², and a TRP channel, TRPM4³³, have been reported to be conjugated and regulated by SUMO, suggesting that SUMOylation may be a common mechanism in functional regulation of ion channels. The regulation by SUMO has been shown to control membrane trafficking, synaptic functions^{27,30}, and more recently nociception³⁴. In the latest case, SUMOylation of a microtubule-binding protein, CRMP2, was shown to be required for proper subcellular localization of the sodium channel subtype, Nav1.7, and critically involved in neuropathic pain³⁵.

The SUMOylation state of a protein is determined by the balance between SUMOylation and deSUMOylation. Whereas SUMOylation is dynamically regulated by activity-dependent redistribution of SUMOylation machinery^{28,36,37}, it is rapidly reversed by the isopeptidase activity of SUMO/sentrin-specific proteases (SENPs), which strongly influences the conjugation/deconjugation balance of SUMO targeted proteins³⁸. Here, we investigated the role of SUMOylation and deSUMOylation in inflammatory thermal hyperalgesia. We show that peripheral inflammation by carrageena injection of mouse hindpaws enhances protein SUMOylation in DRG neurons and conditional deletion of the SENP1 gene in primary somatosensory neurons exacerbated thermal hyperalgesia in both carrageenan- and Complete Freund's adjuvant (CFA)-induced inflammation models. We further identified a lysine residue at

the C-terminus of TRPV1 (K822) to be SUMOylated by SUMO1 and deSUMOylated by SENP1, which when mutated to Arg, not only failed to exhibit a SUMO1-induced channel sensitization to heat stimulus *in vitro* but also was unable to rescue inflammatory thermal hyperalgesia *in vivo* when introduced into DRG neurons of the TRPV1 knockout mice.”

5) The authors use quite low dose of capsaicin in the behavioral analysis shown in Figure 3. Capsaicin was shown to evoke neurogenic inflammation and thermal hyperalgesia in higher doses. The authors should state the reasoning for using capsaicin at such low doses.

Response: We thank the reviewer for bringing this out. We used low dose of capsaicin for exactly this reason, *i.e.* to induce nocifensive behaviors without the complication of inflammation (Khoutorsky et al., 2016). This has been clarified in the text.

References:

Khoutorsky, A. *et al* (2016). eIF2a phosphorylation controls thermal nociception. *Proc. Natl. Acad. Sci. U S A* **113**, 11949-11954.

Referee #2:

This is an interesting paper showing that sumoylation of TRPV1 has an effect on temperature gating of the channel and contributes to heat hyperalgesia. The effects shown seem robust and important. The identification of the sumoylated residue on TRPV1 is a very novel modification of the protein and the rescue experiment done using viral vectors and the mutated sumoylated lysine is a powerful experiment. I have a few suggestions to improve the paper.

Response: We thank the referee for her/his positive evaluation of our work, as well as the very informative and constructive comments that have greatly helped us to improve our manuscript. As requested, we have performed new experiments to address the concerns raised and reworked the introduction and discussion sections so that they are easier to read.

1) The effects are only shown in one inflammation model. Does sumoylation of TRPV1 change in other models (e.g CFA or NGF)? How generally applied is this model?

Response: We thank the referee for raising this important point. As suggested by both reviewers 1 and 2, we have added the CFA model to test the involvement of TRPV1 SUMOylation in regulating nociceptive signaling of inflammatory pain.

As shown in new Fig. 3f, mice that received unilateral intraplantar administration of CFA into the left hind paws developed thermal hyperalgesia

at the injected paws by 2 hrs after injection, which persisted for 7 days throughout the testing period. Thermal hyperalgesia was indicated by a significant reduction in the paw withdrawal latency (PWL) in response to an infrared heat stimulation post-CFA when compared with the contralateral side hindpaw (Fig. 3f). Bonferroni post test showed that sensory thresholds at the ipsilateral side of the CFA-treated mice were significantly different from the contralateral side at all the time points. Importantly, the SENP1 cKO mice exhibited a stronger reduction of PWL than their control littermates, reaching the lowest value of 2.32 ± 0.14 s at 1 day after CFA injection, which remained lower than control throughout the 7 days test period (Fig. 3f). For comparison, the control mice displayed a lowest PWL of 3.32 ± 0.27 s (Fig. 3f). Therefore, the reduced ability to regulate protein SUMOylation and deSUMOylation in DRG neurons because of the SENP1 deficiency exacerbated inflammatory thermal hyperalgesia in both the carrageenan and CFA models.

2) The discussion is really quite long. I found it to be overly verbose, in particular the idea of gating for phosphorylation, which is highly speculative.

Response: We thank the referee for raising this important point. We have extensively revised the Discussion by making it more concise and clearer and moving the part on phosphorylation to Results.

3) the data in DRG with carrageenan in the paw in figure 1 is interesting but the TRPV1 population that must lead to thermal hyperalgesia at this early time point is in the nerve terminals in the paw and not in the DRG cell bodies. Can the authors provide evidence that there is an increase in SUMO modifications at the site of inflammation at this early time point?

Response: We thank the referee for bringing up this interesting point. We agree that the site of action is at the nerve terminals where inflammation occurs. However, a number of technical difficulties have precluded us from accomplishing these experiments. First, nerve terminals represent only a very small portion of the tissue mass at the inflamed sites. The overall SUMOylation levels of the inflamed tissue therefore reflect mainly the status of non-neural tissues. We have compared the levels of SUMOylated proteins in lysates prepared from carrageenan-injected and uninjected paws (Reviewer Fig. 2), but only detected a small trend of decrease in SUMOylation. However, because of the presence of overwhelming amount of surrounding non-neural tissues, we cannot ascertain the SUMO situation in the nerve terminals.

Second, the detection of SUMOylated TRPV1 in the nerve terminals by IP-Western (as shown in Fig. 4c for DRG) is hampered by the strong glycosylation of this protein. Unlike TRPV1 proteins found in the DRG soma, which are minimally glycosylated (Kedei et al., 2001; Xing et al., 2012), those in sensory nerve fibers, and likely at the nerve terminals as well, bear extensive glycosylation, shifting the TRPV1 bands to ~125-135 kD range (a

good example that compares TRPV1 bands in DRG soma vs. sciatic nerves can be found in Fig. 2B of Xing et al., 2012). SUMO tag has an expected molecular weight of 12 kD and migrates at ~15-18 kD. TRPV1 is predicted to be approximately 100.7 kD. Therefore, the SUMOylated TRPV1 would be approximately 120 kD, which is masked by the glycosylated TRPV1. Therefore, our attempt to repeat the IP-western experiments shown for DRG using lysates from inflamed and uninflamed paws did not yield interpretable results (Reviewer Fig. 3).

References:

Kedei N, Szabo T, Lile JD, Treanor JJ, Olah Z, Iadarola MJ & Blumberg PM (2001) Analysis of the native quaternary structure of vanilloid receptor1. J. Biol. Chem. **276**, 28613-28619.

Xing BM, Yang YR, Du JX, Chen HJ, Qi C, Huang ZH, Zhang Y & Wang Y (2012) Cyclin-dependent kinase 5 controls TRPV1 membrane trafficking and the heat sensitivity of nociceptors through KIF13B. J Neurosci. **32**, 14709-14721.

4) I am a little confused on the reasoning in the SENP1 cKO experiment. Since the cKO of SENP1 increases global SUMO1 conjugation in the DRG, why does this not induce thermal hyperalgesia by itself, especially considering that this is a conditional mutation. Does TRPV1 SUMO1 conjugation only change with inflammation while many other proteins change simply by cKO of SENP1? Figure 4 seems to argue against this idea. Is there a location-specific change with more distal changes in nerve terminals not occurring with cKO of SENP1?

Response: Thanks for referee for raising these important and interesting points. A similar question was raised by Reviewer 1.

SUMOylation is a highly dynamic and spatially defined process that regulates multiple cellular events and requires the coordinated activities of the SUMOylation machinery: activating enzyme (E1), conjugating enzyme (E2), SUMO ligases (E3) and deSUMOylating isopeptidases that cooperate with SUMO-targeted ubiquitin ligases. Because of its dynamic nature, SUMOylation is difficult to detect. Typically, only a very small portion of the substrate protein is SUMOylated at any given moment. In addition, there is not a known ligand or cue (either endogenous or exogenous) that can dramatically boost SUMOylation levels, making it hard to deduce the precise cellular mechanism(s) that initiate SUMOylation. Despite these, it has been previously reported that inflammatory mediators increase SUMOylation of retinoid X receptor α , a critical heterodimeric partner of PXR (Schneider Aguirre and Karpen, 2013), suggesting that some constituents of inflammation can enhance SUMOylation. Thus, the balance between SUMOylation and deSUMOylation can be shifted towards SUMOylation under inflammation. Our data suggest that the lack of SENP1 in sensory neurons further exacerbates

such a shift, leading to enhanced thermal hyperalgesia. Notably, the enhanced SUMOylation in DRG from SENP1-cKO animals under basal conditions (Fig. 2c) is not as great as the carrageenan-induced increase (Fig. 1e), supporting that inflammation promotes SUMOylation in DRG neurons. Therefore, there are at least two reasons for the lack of spontaneous pain bouts in naïve SENP1-cKO animals. First, SUMOylation did not reach high enough levels under the basal conditions to trigger pain in the cKO mice. Second, other consequences of inflammation, e.g. already decreased temperature threshold of TRPV1 activation due to a number of mechanisms associated with inflammation, such as acidic pH, PIP₂ breakdown and changes in phosphorylation, etc., work in concert with SUMOylation to support thermal hyperalgesia. The latter would argue that enhanced SUMOylation alone in DRG neurons, with the few °C drop in temperature threshold of TRPV1 (Fig. 6i-l), may be insufficient to cause pain. We have revised Fig. 6 and Supplemental Fig. 8 and the text of Discussion to include these points (Page 17, line 24 to Page 18, line 19).

References:

Schneider Aguirre R & Karpen SJ (2013). Inflammatory mediators increase SUMOylation of retinoid X receptor α in a c-Jun N-terminal kinase-dependent manner in human hepatocellular carcinoma cells. *Mol Pharmacol.* **84**, 218-26.

5) in the discussion, allodynia may not be best explained by sensitization of nociceptors so the authors should be cautious with their language there.

Response: We thank the referee for this suggestion. In our effort to shorten the Discussion, as suggested by the reviewer, we have deleted this sentence.

Referee #3:

The paper by Li and co-workers identifies SUMOylation as a novel post-translational modification of the TRPV1 channel and describes a role for SUMOylation of TRPV1 at lysine residue 822 in the development of thermal hyperalgesia in an inflammatory state. An outstanding finding is the demonstration that TRPV1 knockout mice, which normally do not develop inflammatory thermal hyperalgesia, become hyperalgesia upon “add-back” of a virus overexpressing TRPV1 but fail to develop hyperalgesia when a SUMOylation-deficient TRPV1 channel is added into these mice. The novel findings are based on in vitro and in vivo studies using cell lines, sensory neurons and conditional knockout mice, which collectively support the main findings of the paper. Whilst the behavioral and electrophysiological studies are mostly of good quality and high rigor, the biochemistry is lacking important controls and additional conditions to support the conclusions. Further, the potential cross-talk between other modifications (glycosylation, phosphorylation) of TRPV1

and SUMOylation of TRPV1 is only superficially addressed and needs further development. The statistics and the references cited are appropriate, although specific examples of SUMOylation would be preferred over general reviews.

Response: We appreciate the referee for his/her sincere criticisms, insightful and thoughtful suggestions that have helped us clarify and strengthen our main findings with new experimental data.

Specific comments and suggestions for improvement:

1. Abstract: the statement that SUMO1 and SENP1 are the major regulators of protein SUMOylation is an over-interpretation and should be restated.

Response: We agree with the referee on this important point. As suggested, we have toned-down the writing. The revised manuscript reads as follows: “We report here that SUMO1 and SENP1, regulators of protein SUMOylation, are expressed in primary sensory neurons of mouse dorsal root ganglia (DRG).”

2. As written, the Introduction is vague. For example, instead of using several reviews to cite the importance of SUMOylation in the nervous system – none of which pertain to the pain pathway – perhaps the authors could instead give examples of how, in the nociceptive pathway, protein targets of SUMOylation are involved.

Response: We have revised introduction extensively to make it more concise and clearer. The references are changed to original research papers instead of reviewers.

With respect to the importance of SUMOylation in the nociceptive pathway, however, there was literally no report on whether SUMOylation is involved in pain pathway at the time of our original submission. Therefore, our study likely represents one of the first investigations on SUMO regulation of nociceptive pathway. However, during the revision of our manuscript, a paper appeared showing that CRMP2 SUMOylation is required for proper localization of Na_v1.7, and an increase in CRMP2 SUMOylation during neuropathic pain plays a role in driving nociceptive behaviors (Moutal et al., 2017). We have now revised introduction to reference this recent finding.

References:

Moutal A, Dustrude ET, Largent-Milnes TM, Vanderah TW, Khanna M, Khanna R (2017) Blocking CRMP2 SUMOylation reverses neuropathic pain. Mol Psychiatry. doi: 10.1038/mp.2017.117.

3. Why is SUMOylation increased in carrageenan-treated tissues (Fig. 1d, e)? What are some potential reasons that this might happen? And how might this be relevant?

Response: Similar questions were raised by Reviewers 1 and 2. Basically, SUMOylation is a highly dynamic and spatially defined process that regulates multiple cellular events and requires the coordinated activities of the SUMOylation machinery: activating enzyme (E1), conjugating enzyme (E2), SUMO ligases (E3) and deSUMOylating isopeptidases that cooperate with SUMO-targeted ubiquitin ligases. Because of its dynamic nature, SUMOylation is difficult to detect. Typically, only a very small portion of the substrate protein is SUMOylated at any given moment. In addition, there is not a known ligand or cue (either endogenous or exogenous) that can dramatically boost SUMOylation levels, making it hard to deduce the precise cellular mechanism(s) that initiate SUMOylation. Despite these, it has been previously reported that inflammatory mediators increase SUMOylation of retinoid X receptor α , a critical heterodimeric partner of PXR (Schneider Aguirre and Karpen, 2013), suggesting that some constituents of inflammation can enhance SUMOylation. Thus, the balance between SUMOylation and deSUMOylation can be shifted towards SUMOylation under inflammation. Our data suggest that the lack of SENP1 in sensory neurons further exacerbates such a shift, leading to enhanced thermal hyperalgesia. Notably, the enhanced SUMOylation in DRG from SENP1-cKO animals under basal conditions (Fig. 2c) is not as great as the carrageenan-induced increase (Fig. 1e), supporting that inflammation promotes SUMOylation in DRG neurons.

References:

Schneider Aguirre R & Karpen SJ (2013) Inflammatory mediators increase SUMOylation of retinoid X receptor α in a c-Jun N-terminal kinase-dependent manner in human hepatocellular carcinoma cells. *Mol Pharmacol.* **84**, 218-226.

Why are there no SUMO1-conjugated proteins in the region between 40 to 90 kD in the blots shown in panels d and e of Fig 1? Typically in a cell, there is a high number of proteins of sizes between 40 to 90 kD.

Response: We concur with the reviewer on this observation. In these experiments, we used a pan SUMO-1 Antibody #4930 (Cell Signaling Technology, CST), which is described to be able to detect recombinant SUMO-1 and endogenous SUMOylated proteins. Consistent with the example provided by the manufacturer (<https://www.cellsignal.com/products/primary-antibodies/sumo-1-c9h1-rabbit-mab/4940>), we typically saw a major band of ~90 kD and a minor band of ~50 kD (cut off in most blots, but see Fig. 2c) when using this antibody. Higher molecular weight bands (or smears) are also typical. Intermediate size bands (e.g. 40-90 kD) will show up upon longer exposure.

4. The data in Fig. 4f supporting lys 822 contributing to TRPV1 SUMOylation are convincing. However, the data using native conditions to prove SUMOylation of TRPV1 need additional controls in order to conclude that TRPV1 is being SUMOylated. In Fig 4a, how do the authors conclude that the band at ~170 kD in the SUMO1 blot (top blot) is actually TRPV1 and not another protein of similar size? From the cell lysates shown below, it is clear that there are several proteins in this vicinity. In Fig. 4c, d and f, there are no negative controls for the IP experiments. The authors ought to repeat the experiments immunoprecipitating TRPV1 in denaturing conditions and probing with SUMO1 antibody to convincingly demonstrate that TRPV1 is SUMOylated. Importantly, in all of figure 4 panels, quantification is missing and should be provided. Individual samples values would be useful to report as levels of SUMO seems to be different between samples within the same experiment (Figure 4b).

Response:The estimated size of TRPV1 containing a GFP-SUMO1 tag is ~160 (please see our response to Reviewer 1, Question 3 for detailed discussion). In Fig. 4a, this band was only detected in cells that coexpressed TRPV1 and GFP-SUMO1 and it was removed by the overexpression of SENP1 but not the functional deficient mutant of SENP1. Cells expressing TRPV1 or GFP-SUMO1 alone also served as negative controls for these experiments. To address the reviewer's other questions, we have performed multiple new experiments as follows:

First, as requested, we have repeated IP experiments to prove SUMOylation of TRPV1 under denaturing conditions in transfected cell and tissue homogenates from wild type and SENP1 cKO mice, which is consistent with our previous data from non-denaturing conditions We have included the data in the revised Fig. 4a, b, c, e; Fig. 8a-d; Supplemental Figs. 1, 2 and 3).

Second, as requested, we have added negative control of IP experiments, undertaken further quantification and included quantification results in Fig. 4c.

5. The recordings in Figures 5 and 6 are of good quality but are missing expression of Ubc9. This is critical, as in our experience HEK 293 cells do not express adequate machinery for SUMOylation. Therefore, the experiments in Fig. 5 and 6 (particularly TRPV+SUMO1 and TRPV1-K822R+SUMO1 conditions) must be redone in the presence of over-expressed Ubc9. Note that the authors used an overexpression strategy in Fig. 4f to boost SUMOylation. The authors must also explain and justify their use of CHO-K1 cells for biochemistry and HEK293T cells for electrophysiology. Results need to be reproduced in DRG neurons.

Response: We thank the reviewer for the excellent suggestion. We have performed several additional experiments to address the questions raised.

First, as requested, we have repeated the electrophysiological experiments using cells that over-expressed Ubc9. The new results, now presented in the revised Figs. 5, 6 and Supplemental Figs. 5-7, are consistent with our previous observations that SUMO1 modification of TRPV1 does not alter the sensitivity of TRPV1 to capsaicin, protons, and voltage, with or without the co-expression of Ubc9. However, the co-expression of SUMO1 still significantly lowered the temperature coefficient (Q_{10}) and temperature threshold (T_m) of TRPV1 activation in the presence of over-expressed Ubc9 (Figs. 5, 6 and Supplemental Figs. 5-7).

Second, the switch between HEK293 and CHO-K1 cells was a result of collaboration between the two main laboratories; both are deeply involved in the project but each has their preferred systems to work with. As requested, we have performed additional biochemistry experiments in HEK293 cells and electrophysiology experiments in CHO-K1 cells. The new results reproduced the previous biochemical and electrophysiological findings. These data are included in the revised Figs. 5, 6 and Supplemental Figs. 2, 5.

Finally, as requested by the reviewer, we have carried out additional electrophysiological experiments to evaluate temperature gating of TRPV1 in DRG neurons prepared from wild type and SENP1-cKO mice and included the data in the revised Fig. 6 and Supplemental Fig. 9. The results show that the knockout of SENP1 from DRG neurons selectively affected temperature gating of TRPV1 without affecting other mechanisms of channel activation, supporting the conclusion drawn from the heterologous systems.

6. Fig 7 contains of the two most interesting features of this work: behavioral demonstration of the importance of SUMOylation. It would be important to evaluate behavior in wild type mice following the introduction of the TRPV1 K822R SUMO-mutant as this may reveal whether the loss of SUMOylation is a dominant negative phenotype. The temperature experiments are very well done and highlight the other salient feature of this paper: the ~7 C shift in thermo sensation. Again, as above, at least for the TRPV+SUMO1 and TRPV1-K822R+SUMO1 conditions, the experiments ought to be redone in the presence of over-expressed Ubc9. Results need to be reproduced in DRG neurons.

Response: We thank the reviewer for the excellent suggestions. The second and third parts of this question have been answered in our responses to reviewer's questions 4 and 5 (see above). For the first part, we have performed the suggested behavioral experiment following viral expression of the mutant TRPV1 K822R into the DRG of wild type mice. We saw a moderate, but nevertheless significant, decrease in sensitization to heat stimulation compared to the wild type controls in response to carrageenan-induced inflammation at the hind paw (Fig. 7f). Therefore, the K822R mutant indeed has some weak dominant-negative effect on the development of inflammatory

thermal hyperalgesia.

7. Is TRPV1 phosphorylation changing the SUMOylation status? Which modification is more or most important? The cross-talk between modifications is not sufficiently investigated. The data in Supplementary Fig. 4a, for example, shows that there is some change in PKC phosphorylation in the TRPV1 mutant but no data is provided quantifying it. The cross-talk between modifications is an interesting aspect of regulation of proteins and may be the case here for TRPV1 channels, too.

Response: Reviewer 1 raised similar issue (Reviewer 1, Question 5). We thank both reviewers for raising this important point. To address this question, we have performed the following experiments.

First, we have performed new experiments to examine TRPV1 phosphorylation in the absence and presence of SUMOylation. The new results are incorporated into previous Supplemental Fig. 4 and the figure has been moved to the main part of the manuscript as revised Fig. 8.

Second, as suggested by reviewer 1, we have used the viral delivering system to introduce TRPV1(S502A/S800A) with and without the K822R mutation to DRG neurons of the *Trpv1* null mice and then assessed inflammatory thermal hyperalgesia in the carrageenan edema model. The TRPV1(S502A/S800A)-injected mice exhibited similar PWL in response to radiant heat as the wild type TRPV1-injected ones (Fig. 8e), while the combination with K822R, TRPV1(S502A/S800A/K822R), eliminated such a response (Fig. 8e). These results suggest that TRPV1 SUMOylation is essential for the development of inflammatory thermal hyperalgesia, and the response is not dependent upon TRPV1 phosphorylation, at least at S502 and S800. In the revised Fig. 8c and d demonstrate that TRPV1 SUMOylation promotes its phosphorylation.

Minor suggestions:

1. In the manuscript sumoylation/desumoylation should be written SUMOylation/deSUMOylation since SUMO is an acronym.

Response: We have substituted the word "sumoylation/desumoylation" with "SUMOylation/deSUMOylation" throughout the article

2. Product numbers for all materials should be provided.

Response: We have included these numbers for all materials in the revised manuscript as requested.

3. Details on virus production and titration are missing.

Response: We have added detailed information on virus production and titration in Materials and Methods section of the revised manuscript as requested.

4. Need to control for loading in SUMO IP experiments

Response: Thank you for the observation. We have included the information in Fig 4c.

5. Please provide a neuronal marker staining for figure 7C.

Response: We thank the referee for pointing this out and have now included a neuronal marker staining for Fig. 7c.

6. No negative control for figure sup 1

Response: We thank the referee for pointing this out and have now included a negative control for previous figure sup 1 (now revised Supplemental Fig. 3a).

7. Supplementary Figure 2, flag sumo is not in the right place. Authors are expecting a higher shift than the experiment is suggesting. Also, no negative control provided.

Response: We thank the referee for pointing this out. We have corrected these and added the negative control.

8. No negative control for surface expression provide in Supplementary figure 3.

Response: We thank the referee for pointing this out and have now included a negative control for previous Supplementary figure 3 (now Supplemental Fig. 4a).

9. A general suggestion in some of the biochemistry experiments would be to use the GFP antibody instead of the SUMO1 antibody.

Response: In accordance with the reviewer's comment, we have carried out this experiment again using the GFP antibody instead of the SUMO1 antibody. We have replaced the original figure with the new data in the revised Fig. 4.

Additional figure for the reviewers

Reviewer Fig.1 At 1 h after intraplantar injection of carrageenan into the left hindpaws (Ipsi), L3-L4 DRG from both sides and spinal cord were dissected for western blotting for Ubc9, SENP1 and GAPDH.

Reviewer Fig.2 At 0.5 and 1 h after intraplantar injection of carrageenan into the left hindpaw, L3-L4 DRG from both sides and nerve ending were dissected for western blotting for SUMO1 and GAPDH.

Reviewer Fig.3 At 1 h after intraplantar injection of carrageenan into the left hindpaw, nerve ending tissues from both sides were lysed under denature condition. Then cell lysates were IP with anti-TRPV1 antibody and analyzed by IB using anti-TRPV1 or anti-SUMO1. Red arrow: TRPV1 band; red rectangle: TRPV1 SUMO band.

Reviewer Fig.4 DRG tissues were lysed under denature condition. Then cell lysates were incubated with PNGase F for 1 h at 37 °C, then IP with anti-TRPV1 antibody and analyzed by IB using anti-TRPV1 or anti-SUMO1. Red arrow: TRPV1 band; red rectangle: TRPV1 SUMO band.

REVIEWERS' COMMENTS:

Reviewer #1 (Remarks to the Author):

The authors have fully addressed all of my concerns. Importantly, the authors added the CFA behavioral test, and demonstrated that indeed their initial results concerning the role of sumoylation of TRPV1 during inflammation is valid in rodents. The authors re-wrote several parts of the manuscript, and now the manuscript is much stronger and clearer. Thus, I feel that at this point the manuscript is ready for publication, mainly due to its novelty and strong scientific evidence.

Reviewer #2 (Remarks to the Author):

The authors have made substantial edits to the paper and provided a good deal of new data that is very convincing. While the scope of the findings is limited (thermal hyperalgesia only) the principle findings are quite novel and will likely have an impact on the field.

Reviewer #3 (Remarks to the Author):

The authors have addressed my concerns with elegantly performed new experiments, focused introduction, and improved discussion. Overall, the sum of these changes have significantly improved this research article including more focused writing and discussion.

Minor concerns:

- Line 100 to 103 the citation is incorrect - it should be 35 (Dustrude et al., 2016)
- Line 111: typo on the word carrageenan
- Line 837: delete "previously described"
- Figure 7f: the black line is not labeled
- Figure 8: The font for the bar graphs is too small and will be difficult to read in the published version of the paper.
- The authors state that they added a citation by Moutal et al 2017 to the introduction but this is not listed in the Intro or t References sections

Rajesh Khanna, PhD
University of Arizona

Authors' Responses to the Reviewers' Comments:

Reviewer #1 (reviewers' comments in *italic*):

The authors have fully addressed all of my concerns. Importantly, the authors added the CFA behavioral test, and demonstrated that indeed their initial results concerning the role of sumoylation of TRPV1 during inflammation is valid in rodents. The authors re-wrote several parts of the manuscript, and now the manuscript is much stronger and clearer. Thus, I feel that at this point the manuscript is ready for publication, mainly due to its novelty and strong scientific evidence.

We are grateful to this reviewer for his/her enthusiastic support.

Referee #2:

The authors have made substantial edits to the paper and provided a good deal of new data that is very convincing. While the scope of the findings is limited (thermal hyperalgesia only) the principle findings are quite novel and will likely have an impact on the field.

We thank the reviewer for the support and recommendation to accept our paper.

Reviewer #3:

The authors have addressed my concerns with elegantly performed new experiments, focused introduction, and improved discussion. Overall, the sum of these changes have significantly improved this research article including more focused writing and discussion.

We thank the reviewer for his positive comments on our work and careful evaluation of the manuscript. We also thank him for his critical comments and insightful suggestions that have helped us clarify and strengthen our main findings.

Minor concerns:

- Line 100 to 103 the citation is incorrect - it should be 35 (Dustrude et al., 2016)

Response: We thank the reviewer for catching this mistake. The corresponding text has been revised accordingly.

- Line 111: typo on the word carrageenan

Response: We thank the reviewer for pointing this out. We have corrected the typo in the manuscript.

- Line 837: delete "previously described"

Response: This has been corrected as suggested.

- Figure 7f: the black line is not labeled

Response: We have updated the missing information in the revised manuscript.

- Figure 8: The font for the bar graphs is too small and will be difficult to read in the published version of the paper.

Response: We thank the reviewer for the suggestion. We have increased the font size of bar graphs in the revised Fig. 10.

- The authors state that they added a citation by Moutal et al 2017 to the introduction but this is not listed in the Intro or t References sections

Response: We thank the reviewer for catching this. The citation is now included in the revised manuscript.

We hope that the revised manuscript with minor text edits is satisfactory and that you will formally accept this version for publication in Nature Communications.